

# A taxonomic revision of the south-eastern dragon lizards of the *Smaug warreni* (Boulenger) species complex in southern Africa, with the description of a new species (Squamata: Cordylidae)

Michael F. Bates[1,2,*] and Edward L. Stanley[3,*]

[1] Department of Herpetology, National Museum, Bloemfontein, South Africa
[2] Department of Zoology and Entomology, University of the Free State, Bloemfontein, South Africa
[3] Department of Herpetology, Florida Museum of Natural History, Gainesville, FL, USA
* These authors contributed equally to this work.

Corresponding author
Michael F. Bates, herp@nasmus.co.za

## ABSTRACT

A recent multilocus molecular phylogeny of the large dragon lizards of the genus *Smaug* Stanley et al. (2011) recovered a south-eastern clade of two relatively lightly-armoured, geographically-proximate species (*Smaug warreni* (Boulenger, 1908) and *S. barbertonensis* (Van Dam, 1921)). Unexpectedly, *S. barbertonensis* was found to be paraphyletic, with individuals sampled from northern Eswatini (formerly Swaziland) being more closely related to *S. warreni* than to *S. barbertonensis* from the type locality of Barberton in Mpumalanga Province, South Africa. Examination of voucher specimens used for the molecular analysis, as well as most other available museum material of the three lineages, indicated that the 'Eswatini' lineage—including populations in a small area on the northern Eswatini–Mpumalanga border, and northern KwaZulu–Natal Province in South Africa—was readily distinguishable from *S. barbertonensis* sensu stricto (and *S. warreni*) by its unique dorsal, lateral and ventral colour patterns. In order to further assess the taxonomic status of the three populations, a detailed morphological analysis was conducted. Multivariate analyses of scale counts and body dimensions indicated that the 'Eswatini' lineage and *S. warreni* were most similar. In particular, *S. barbertonensis* differed from the other two lineages by its generally lower numbers of transverse rows of dorsal scales, and a relatively wider head. High resolution Computed Tomography also revealed differences in cranial osteology between specimens from the three lineages. The 'Eswatini' lineage is described here as a new species, *Smaug swazicus* sp. nov., representing the ninth known species of dragon lizard. The new species appears to be near-endemic to Eswatini, with about 90% of its range located there. Our study indicates that *S. barbertonensis* sensu stricto is therefore a South African endemic restricted to an altitudinal band of about 300 m in the Barberton–Nelspruit–Khandizwe area of eastern Mpumalanga Province, while *S. warreni* is endemic to the narrow Lebombo Mountain range of South Africa, Eswatini and Mozambique. We present a detailed distribution map for the three species, and a revised diagnostic key to the genus *Smaug*.

## INTRODUCTION

The Cordylidae consists of two subfamilies, Cordylinae (nine genera, 52 species) and Platysaurinae (one genus, 16 species), and is the only lizard family endemic to the mainland of Africa (*Stanley et al., 2011*, *2016*; *Bates et al., 2014*; *Reissig, 2014*; *Whiting et al., 2015*; *Marques et al., 2019*; *Uetz, Freed & Hosek, 2019*). Until recently, only four genera (*Cordylus* Laurenti, 1768, *Chamaesaura* Schneider, 1801, *Pseudocordylus* Smith, 1838, *Platysaurus* Smith, 1844) were recognised in the family, but *Stanley et al. (2011)* erected five new genera (*Smaug, Ninurta, Ouroborus, Karusasaura, Namazonurus*) and resurrected *Hemicordylus* Smith, 1838. The genus *Smaug* consists of eight species, of which the large and only terrestrial form, *S. giganteus* (*Smith, 1844*), is genetically highly divergent (*Stanley et al., 2011*; *Stanley & Bates, 2014*). The other seven species had been treated as the 'Cordylus warreni' (*Boulenger, 1908*) species complex (*Branch, 1988*; *Jacobsen, 1989*).

Members of the 'Cordylus warreni' species complex (*S. warreni, S. barbertonensis* (*Van Dam, 1921*), *S. depressus* (*FitzSimons, 1930*), *S. breyeri* (*Van Dam, 1921*), *S. vandami* (*FitzSimons, 1930*), *S. mossambicus* (*FitzSimons, 1958*) and *S. regius* (*Broadley, 1962*)) are large, robust and spinose girdled lizards (family Cordylidae) restricted to high-elevation regions of the north-eastern provinces of South Africa and Eswatini (also spelled 'eSwatini', formerly Swaziland), and the highlands of eastern Zimbabwe and adjacent Mozambique (*Branch, 1998*). Like most girdled lizards, members of this complex are strictly rupicolous, inhabiting deep, horizontal or gently sloping crevices, often in shaded rocky outcrops (*Jacobsen, 1989*; *Stanley & Bates, 2014*). Due to their reliance on deep crevices they appear to be relatively substrate-specific, occurring in partially-vegetated boulder fields on gentle slopes.

The seven currently recognised taxa in the *S. warreni* complex (as defined above) are allopatric, occurring on separate mountain chains, and are distinguishable on the basis of differences in scalation and colour pattern (*Jacobsen, 1989*; *Branch, 1998*; *Bates et al., 2014*; *Stanley & Bates, 2014*). Despite these clear diagnoses, the *S. warreni* group has a tortuous taxonomic history (see *Stanley & Bates, 2014*). For example, *FitzSimons (1943)* treated *Cordylus barbertonensis, C. b. depressus* and *C. breyeri* as subspecies of *C. warreni*, retained the subspecies *C. vandami perkoensis* (*FitzSimons, 1930*), and continued to recognise *C. laevigatus* (*FitzSimons, 1933*) as a valid species. Shortly thereafter, *Loveridge (1944)* revised the Cordylidae and treated all seven of the above taxa as subspecies of *Cordylus warreni*. *FitzSimons (1958)* later described *Cordylus warreni mossambicus*, and *Broadley (1962)* described *C. warreni regius*. *Cordylus warreni* was therefore considered a polytypic species with as many as nine subspecies (*Branch, 1988*). *Jacobsen (1989)* subsequently investigated the status of South African populations and on the basis of sympatry between *C. w. vandami* and *C. w. breyeri* at one locality, he recognised the former as a full species. As a result of overlapping morphological character variation (scalation

and colour pattern) he considered *C. w. perkoensis* a junior synonym of *C. vandami*, and *C. w. laevigatus* a junior synonym of *C. w. depressus*. *Branch (1998)* later followed *Jacobsen's (1989)* arrangement for South African and Eswatini taxa, but also treated *C. w. breyeri, C. w. mossambicus* and *C. w. regius* as valid species (without providing reasons). *Broadley (2006)* treated all seven taxa in the 'C. warreni' complex (except *laevigatus* and *perkoensis*) as full species, but he too failed to provide justification for such action.

In a recent multilocus molecular study using three mitochondrial and three nuclear genes, *Stanley et al. (2011)* recovered the genus *Cordylus* as paraphyletic and allocated all members of the *C. warreni* complex, together with the large terrestrial species *C. giganteus*, to a new genus, *Smaug*. A subsequent multilocus molecular phylogeny— using three mitochondrial and eight nuclear genes—that focused on the *S. warreni* complex found that *S. warreni, S. barbertonensis, S. depressus, S. breyeri, S. vandami, S. mossambicus* and *S. regius* are all valid species (*Stanley & Bates, 2014*) (Fig. 1). The latter authors identified a south-eastern clade of three species-level taxa (hereafter referred to as the *S. warreni* species complex), comprising *S. warreni* and two lineages of *S. barbertonensis* from northern Eswatini and Mpumalanga Province, South Africa. The latter taxon was shown to be paraphyletic, with samples from northern Eswatini being more closely related to *S. warreni* than to topotypic *S. barbertonensis*, and with genetic distances between the three lineages of 6–10% for the mitochondrial marker ND2. This led us to hypothesise that diagnosable morphological differences should exist between specimens referable to the three lineages.

*FitzSimons (1943)* had noted regional differences in colouration in specimens of *Cordylus warreni barbertonensis* as follows: "sides of body and tail with vertical barring of yellow", "Lower surfaces brown, with irregularly scattered yellowish spots or short transverse bars" (Barberton, South Africa) vs. "sides of body and tail with series of large yellow spots and narrow dark interspaces", "lower surfaces yellowish-white, with irregular dark brown transverse bars on chest and belly, chin spotted with blackish and throat with vermiculate blackish markings" (Eswatini), but he did not suspect that this indicated separate taxonomic status for the two colour forms. *Jacobsen (1989)* examined 24 specimens of *C. w. barbertonensis* from Mpumalanga and the adjacent northern part of KwaZulu–Natal (formerly part of Transvaal), but did not distinguish different colour patterns.

In the present study it was found that specimens of the two 'S. barbertonensis' lineages had consistently different dorsal, lateral and ventral colour patterns, as well as other morphological differences. Populations from Eswatini and adjacent areas in Mpumalanga and KwaZulu–Natal provinces in South Africa, initially referred to as 'Smaug cf. barbertonensis' in this article, are therefore described here as a new species.

## MATERIALS AND METHODS

### Study area
The study area comprises the South African provinces of Mpumalanga and (northern) KwaZulu–Natal, as well as Eswatini and adjacent parts of southern Mozambique. This area is bounded by latitudes 25°S and 28°S, and longitudes 30°30′E and 32°30′E.

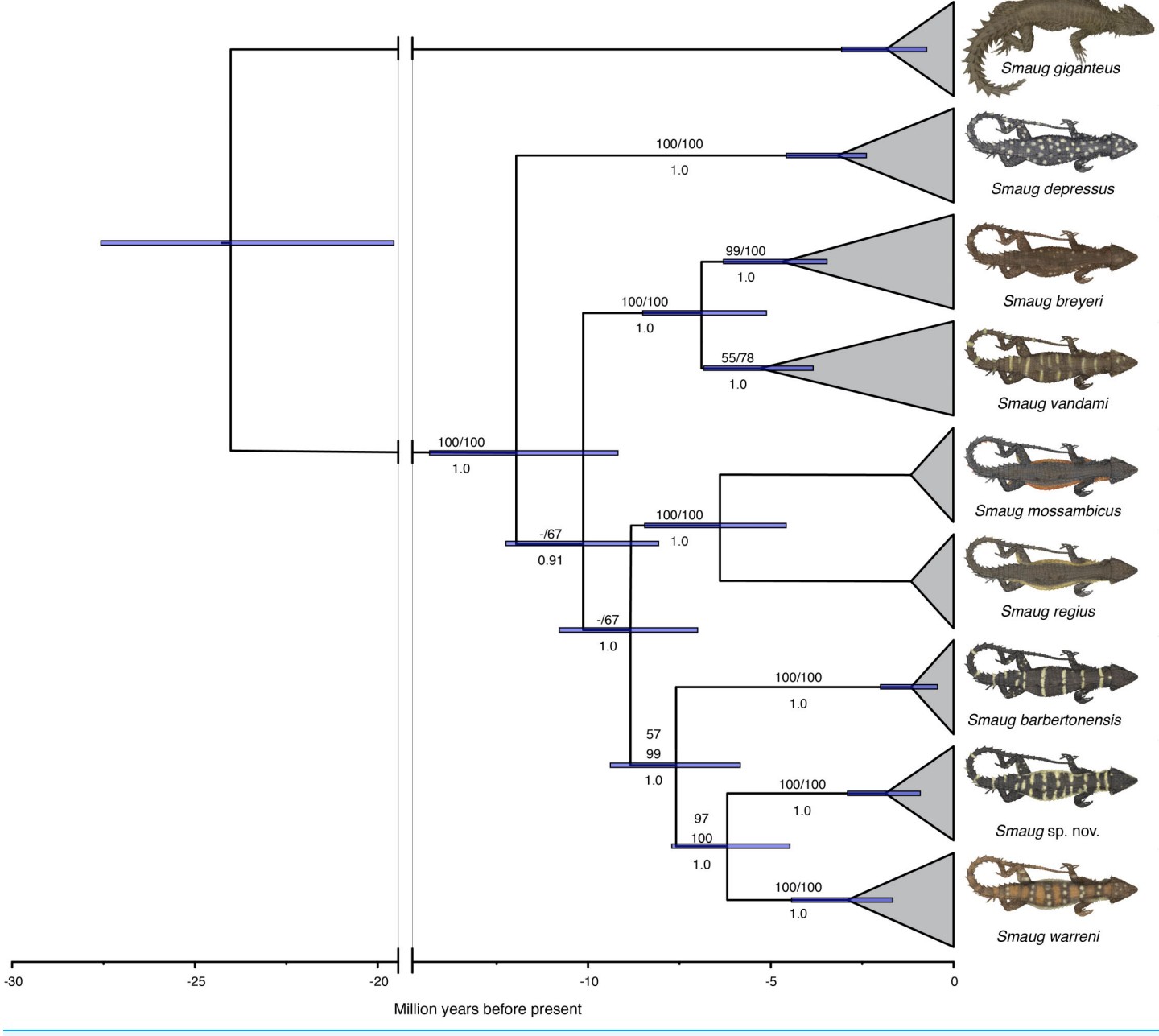

**Figure 1 Time-calibrated phylogram of 11 concatenated nuclear and mitochondrial genes for the genus _Smaug_.** Bootstrap support values (MP/ML) are shown above branches, and posterior probabilities below branches. (Modified from: _Stanley & Bates, 2014_, Fig. 4.) (Image produced by: E.L. Stanley).

## Material examined

All available specimens in the Ditsong National Museum of Natural History, Pretoria (TM) and National Museum, Bloemfontein (NMB) were examined by MFB. Material of _Smaug_ collected during _Jacobsen's (1989)_ survey of the former Transvaal Province and Boycott's (1992) survey of Eswatini was, for the most part, deposited at Ditsong, and this includes the vast majority of museum material identified as _S. warreni_ and

*S. barbertonensis.* Some non-types of the new species referred to below are housed at the American Museum of Natural History, New York (AMNH), Durban Natural Science Museum, Durban (DNSM), and Natural History Museum of Zimbabwe, Bulawayo (NMZB); and a few specimens of *S. warreni* are in the collections of AMNH and NMZB.

When collection co-ordinates (presented as degrees, minutes and in many cases seconds) and/or altitudes (m above sea level) were not available in museum documentation, these were estimated using Google Earth Pro.

In addition to the data presented in this article, comparative data consulted for the diagnoses of species and for the purposes of preparing a diagnostic key (see below) were obtained from NMB and NMZB specimens listed in the appendix in *Mouton et al. (2018)*, and data in *Boulenger (1908)*, *Van Dam (1921)*, *FitzSimons (1930*, *1933*, *1943*, *1958)*, *Loveridge (1944)*, *Broadley (1962*, *1966)*, *De Waal (1978)*, *Jacobsen (1989)*, *Stanley et al. (2011)* and *Mouton et al. (2018)*.

## Ethics approval
This project was approved by the National Museum Bloemfontein Ethics Clearance Committee (NMB ECC 2019/13).

## External morphology
### Measurements
Snout to vent length (SVL) was measured from the tip of the snout to the vent after flattening the specimen on its back; tail length, from vent to tip of tail; using vernier calipers or a millimeter ruler. Head measurements (determined using vernier callipers and unless damaged, taken on the right side): Length, measured from tip of snout to ear opening; width, at widest point at about the level of the posterior borders of the parietals; depth, from middle of posterior sublabial to highest point of posterior parietal.

### Scalation (examined by MFB using a binocular dissecting microscope, mostly a Nikon SMZ 745T; both sides of head examined)
For the most part the morphological characters employed by *FitzSimons (1943)* were used, and in the same way, unless otherwise indicated. To avoid uncertainty, the following scale counts are described in detail: occipitals: large scales behind the posterior parietals, the outermost ones situated directly behind the elongated upper temporals; gular scales (often elongated and in longitudinal rows): counted transversely between posterior sublabials, the first row extending to the anterior end of the posterior sublabial; dorsal scale rows longitudinally: counted across the widest part of the body more-or-less midway between fore- and hindlimbs (scales of the most lateral rows are at least half the width of adjacent enlarged dorsals); dorsal scale rows transversely: counted from the first complete row behind the occipitals to the row that ends immediately anterior to the vent (when followed around to the ventral side; excluding rows less than three-quarters as long as adjacent ones); ventral scale rows longitudinally: counted across the widest part of the body, more-or-less midway between fore- and hindlimbs (lateral ventrals are rectangular or quadrangular, smooth or weakly keeled, flattened and at least half the size and width of

adjacent ventrals); ventral scale rows transversely: counted from the first row (which curves anteriorly) behind the posterior part of the forelimb insertion to the row (which curves posteriorly) immediately in front of the anterior part of the hindlimb insertion (i.e. scale rows between axilla and groin); lamellae under 4th toe of right foot were counted from the first scale entirely or largely (>60%) anterior to the junction between 3rd and 4th toes to the scale behind the claw, and incomplete lamellae (i.e. those that do not extend to either side) were excluded.

### Sexing

Males (>70 mm SVL) were identified by the presence of large femoral pores (usually with waxy plugs of secreted fluid) as well as differentiated femoral scales (generation glands). Females (>70 mm SVL) had minute pin-prick-like femoral pores without waxy plugs, and lacked differentiated femoral scales.

## Osteological data

Osteological data was obtained from representative specimens of the *S. warreni* species complex via High Resolution X-ray Computed Tomography (HRCT). Specimens used were: *S. warreni* NMB R9292, AMNH-R-173381; *S. barbertonensis* NMB R9196 (topotype); *S.* cf. *barbertonensis* NMB R9201 (holotype of new species, see below), AMNH-R-173382. These specimens were scanned using a Phoenix v|tome|x S CT scanner at the American Museum of Natural History's Microscopy and Imaging Facility in New York, and GE Inspection Technologies, LP Technical Solutions Center in San Carlos, CA, or on a Phoenix v|tome|x M at the University of Florida's Nanoscale Research Facility in Gainesville, FL. Each specimen was scanned twice: once to recover the full body, and a second higher resolution scan to focus on the cranial morphology. Current, voltage and detector-time were modified to optimise the greyscale range, and specimens were scanned in sections to maximise resolution (Table S1). Raw data were processed using GE's proprietary datos|x software V.2.3 to produce a series of tomogram images which were then viewed, sectioned, measured and analysed using VG Studio Max 2.2 (Volume Graphics, Heidelberg, Germany). Individual skeletal elements and osteoderms were reconstructed separately for each scan, so as to facilitate osteological analysis. Tomograms and 3D mesh files for all datasets are available online at www.morphosource.org (see Supplemental Data for DOIs).

## Statistical analyses

Univariate analyses of scale counts were conducted using Statistica v. 6. Principal component and linear discriminant analyses were run for three mensural characters (head length, width and height) and 13 meristic characters (supraciliaries, suboculars, supralabials, infralabials, sublabials, occipitals, gulars, dorsal scale rows transversely and longitudinally, ventral scale rows transversely and longitudinally, femoral pores, subdigital lamellae on fourth toe), taken from 72 museum specimens (>70 mm SVL; i.e. juveniles excluded to avoid the effects of ontogenetic growth) (Table S2), using the prcomp and lda commands in R {stats} and {MASS}. When scale counts were made on both sides of the head or on both hindlimbs (see Table S2), a mean value was used for the analyses.

## Species concept and species delimitation

We apply a lineage-based species concept whereby a species is represented by an independently evolving metapopulation lineage (see *Frost & Hillis, 1990*; *De Quieroz, 1998*, *2007*). The genetic distinctness described by *Stanley & Bates (2014)* and morphological characters were the operational criteria for species delimitation. Although *Stanley & Bates (2014)* did not employ coalescent species delimitation analyses, the topological consistency of the mtDNA, nuDNA and combined analyses strongly support the existence of three distinct lineages within *S. warreni* and *S. barbertonensis*.

## Nomenclatural note

The electronic version of this article in Portable Document Format (PDF) will represent a published work according to the International Commission on Zoological Nomenclature (ICZN), and hence the new names contained in the electronic version are effectively published under that Code from the electronic edition alone. This published work and the nomenclatural acts it contains have been registered in ZooBank, the online registration system for the ICZN. The ZooBank LSIDs (Life Science Identifiers) can be resolved and the associated information viewed through any standard web browser by appending the LSID to the prefix http://zoobank.org/. The LSID for this publication is: (urn:lsid:zoobank. org:pub:490BDD66-155F-423F-A4E9-DEAEEB024CC5). The online version of this work is archived and available from the following digital repositories: PeerJ, PubMed Central and CLOCKSS.

## RESULTS

### Character analysis

#### *Dorsal colour pattern*

Figure 2 Specimens from all three clades recovered by *Stanley & Bates (2014)* are distinguishable on the basis of dorsal, lateral and ventral colour patterns. *S. warreni* has a medium to sandy brown (sometimes reddish-brown) dorsum with a series of 5-6 interrupted transverse bands between fore- and hindlimbs, each consisting of white or cream ocelli (spots or blotches) with dark (often black) borders. The dark-edges exaggerate the ocelli, but some specimens have only small pale markings which also lack heavy dark borders. The dorsum of *S. barbertonensis* is medium to dark brown (or even black), usually with 4-5 interrupted bands on the back formed mostly by transversely enlarged pale markings (rather than spots or blotches) with moderately dark edges. *Smaug* cf. *barbertonensis* is similar to the latter form, but there are usually 5–6 bands. However, in *S. barbertonensis* there is almost always a pale spot on the nape immediately posterior to the median occipitals, followed in close proximity by a distinct transverse band. In *S.* cf. *barbertonensis*, the spot on the nape is replaced by a pale band, followed after a distinct gap by another pale band (often divided medially) on the neck with a slightly posteriorly-directed curvature. *S. warreni* also has a pale band behind the occipitals, but the band that follows is seldom curved as in the case of the previous form.

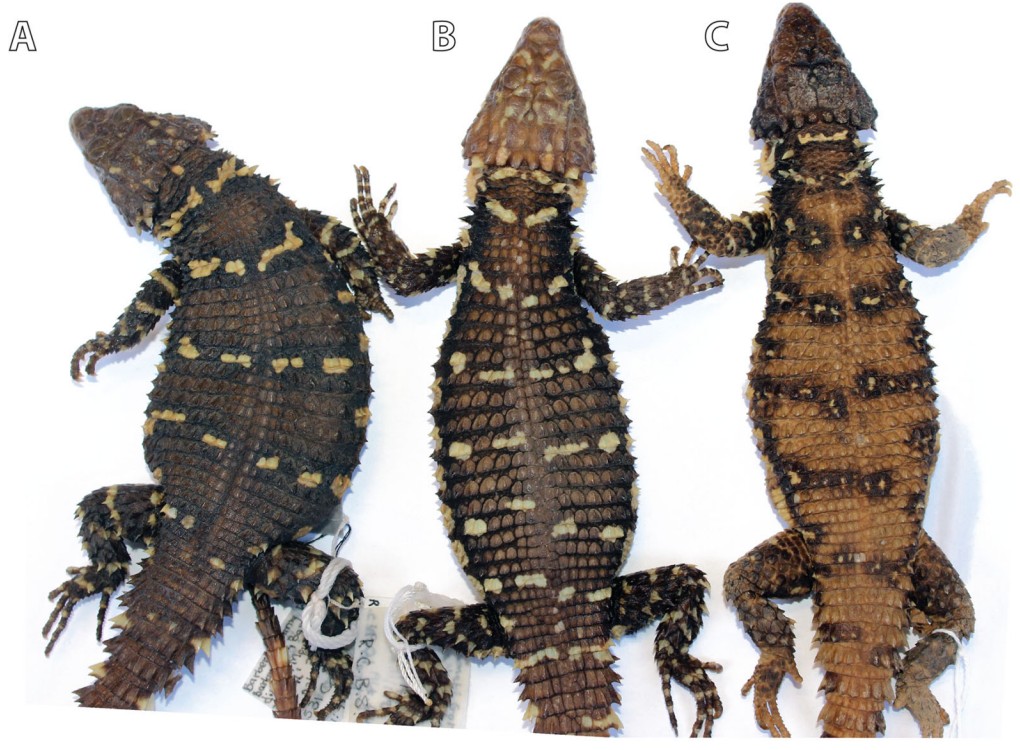

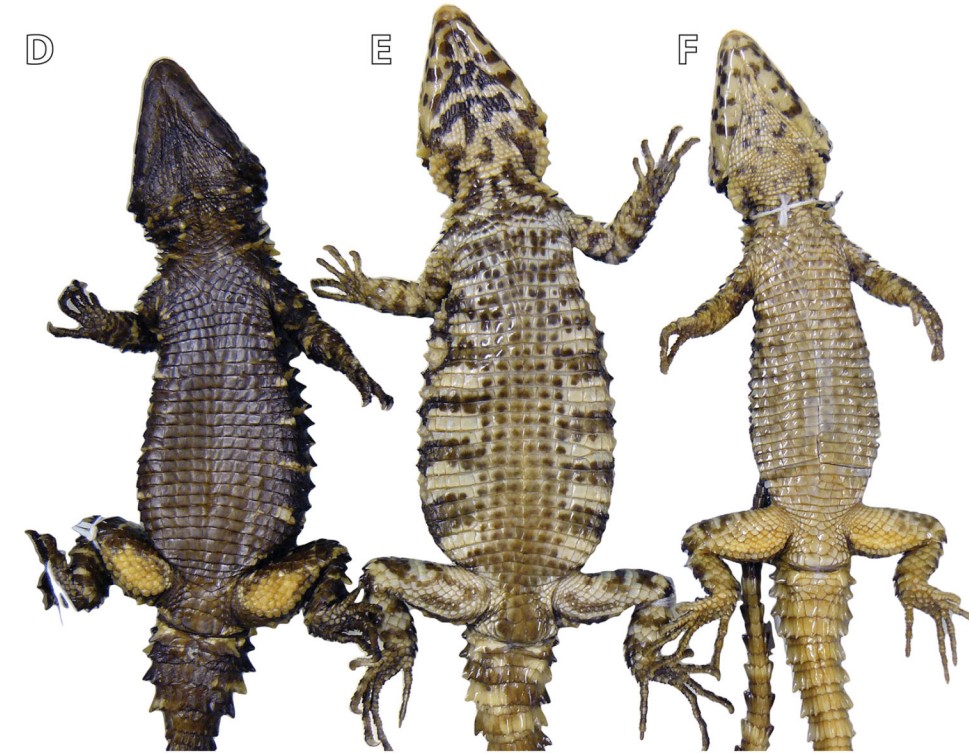

**Figure 2 Differences in colour pattern in the *Smaug warreni* species complex.** From left to right: Dorsal views of (A) *Smaug barbertonensis* (NMB R9196, topotype), (B) *S.* cf. *barbertonensis* (TM 78918, allotype of new species, see below) and (C) *S. warreni* (TM 63567); ventral views of (D) *S. barbertonensis* (TM 55789), (E) *S.* cf. *barbertonensis* (TM 78918) and (F) *S. warreni* (TM 78969) (Photo credits: M.F. Bates).

### Ventral colour pattern

Figure 2: In *S. warreni* the belly is generally white with the centre of each scale pale brown; the throat is usually mostly white with scattered small to medium-sized dark brown spots. In *S. barbertonensis* the belly is almost completely black or dark brown, with only a few pale markings on the sides; the throat is also almost entirely dark, with only occasional pale specks or blotches. In *S.* cf. *barbertonensis* the belly is white with 5–6 broad, dark brown 'cross-bands', interrupted mid-ventrally by six longitudinal rows of brown scales; the throat is white with bold, dark mottling or reticulations (sometimes forming transverse bands; often most of the throat is dark).

### Lateral colour pattern

The flanks of *S. warreni* are often mostly cream with a few dark markings, but may consist of alternating light and dark vertical bands. In *S. barbertonensis* the flanks are primarily dark brown or even black, with a few narrow or moderate cream bands and/or spots/blotches. In contrast, the sides of the body in *S.* cf. *barbertonensis* consist of large cream spots or blotches on a dark background; and in some cases the light patch behind the armpit is elongated (antero-posteriorly).

### Scales at the edges of the ear openings

*Smaug barbertonensis* usually has generally elongated and spinose scales at the anterior edges of the ear openings (especially the central ones), whereas in most cases these scales are short and non-spinose in *S.* cf. *barbertonensis* and *S. warreni*.

### Relative length of occipital scales

In all three forms in the complex there are usually six occipital scales, and the scales of the median pair are shorter and usually smaller than the others (although usually less distinctly so in *S. warreni*). In *S. warreni* the outer occipital is usually of similar size and shape to the adjacent inner occipital, but in *S. barbertonensis* and *S.* cf. *barbertonensis* the outer one is usually shorter and smaller. In *S. warreni* a small median occipital is common.

### Quadrate variation

In *S.* cf. *barbertonensis* the quadrates have a pronounced ridge and concave region at the lateral edge of the *adductor musculus mandibulae* posterior origin, whereas in *S. barbertonensis* and *S. warreni* the quadrates have a less pronounced ridge and a non-concave region (Fig. 3).

### Scale counts

The three forms are similar in terms of scale counts (Table 1), but *S. barbertonensis* usually has lower numbers of transverse dorsal scale rows than *S. warreni* and *S.* cf. *barbertonensis* (28–34 vs. 31–41; Fig. 4A).

### Head width

*Smaug barbertonensis*, when compared to both *S. warreni* and *S.* cf. *barbertonensis*, usually has a wider head relative to snout-vent length (SVL) (Fig. 4B).

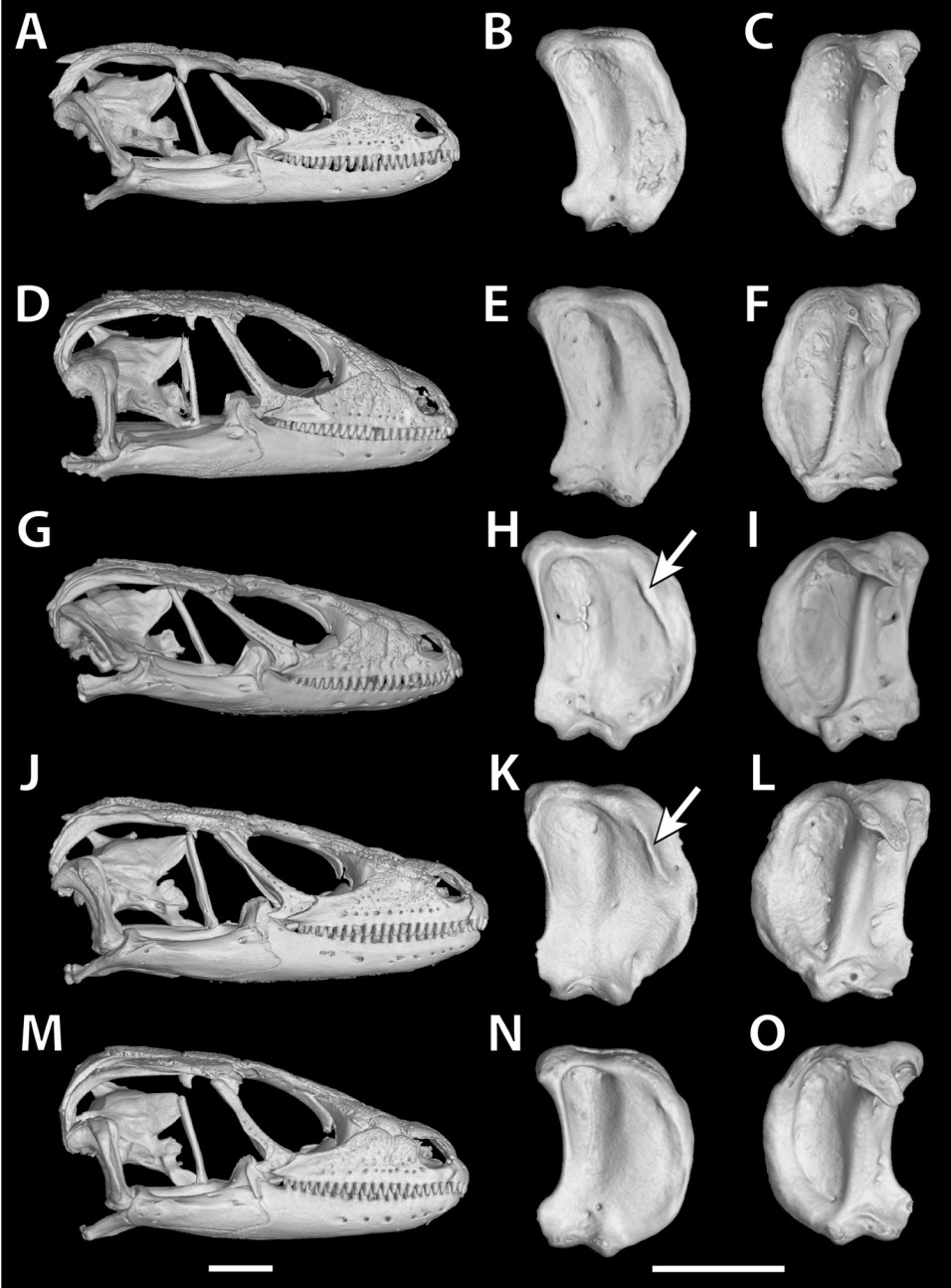

**Figure 3 Quadrate variation in the *Smaug warreni* species complex.** (A–C) *S. warreni* NMB R9292. (D–F) *S. warreni* AMNH-R-173381. (G–I) *S.* cf. *barbertonensis* AMNH-R-173382. (J–L) *S.* cf. *barbertonensis* NMB R9201 (holotype of new species, see below). (M–O) *S. barbertonensis* NMB R9196 (topotype). Diagnostic ridges on the quadrates of *S.* cf. *barbertonensis* are indicated using arrows. Horizontal bar = 5 mm. (Images produced by: E.L. Stanley).

### Spinosity

A recent study by *Mouton et al. (2018)* that investigated the relationship between generation gland morphology and armour in the genus *Smaug* found that those species with multi-layer generation glands (*S. giganteus*, *S. breyeri*, *S. vandami*) had relatively

**Table 1 Comparison of scalation data in the three lineages/populations of the *Smaug warreni* species complex.**

| | Dorsal scale rows longitudinally | Dorsal scale rows transversely | Ventral scale rows longitudinally | Ventral scale rows transversely | Lamellae under 4th toe (usually right) | Femoral pores (per thigh) (males and females) | Differentiated femoral scales in males (per thigh) | Gular scales between posterior sublabials |
|---|---|---|---|---|---|---|---|---|
| *Smaug* cf. *barbertonensis* | 20–26 22.6 ± 1.53 N = 22 | 31–41 34.7 ± 2.37 N = 22 | 14 (12[4]) N = 22 | 23–29 26.3 ± 1.24 N = 21 | 16–19 17.6 ± 0.85 N = 22 | 10–13 11.2 ± 0.90 N = 17 | 9.5–33 23.2 ± 7.25 N = 9 | 22–29 25.7 ± 1.93 N = 20 |
| *Smaug barbertonensis* | 20–24 21.7 ± 1.46 N = 26 | 28–34 30.5 ± 1.48 N = 26 | 14 (16[1]) N = 26 | 25–28 26.4 ± 0.86 N = 26 | 15–19 17.7 ± 1.04 N = 26 | 8.5–11.5 9.9 ± 0.79 N = 20 | 18.5–35.5 25.9 ± 5.16 N = 9 | (20) 23–31 27.0 ± 2.47 N = 26 |
| *Smaug warreni* | 22–28 23.6 ± 1.42 N = 39 | 31–41 35.2 ± 2.11 N = 39 | 14 (12[2], 13[1]) N = 39 | 23–27 25.9 ± 1.00 N = 39 | 15–20 16.8 ± 1.20 N = 39 | 7.5–13 10.4 ± 1.30 N = 31 | 14.5–38 25.7 ± 6.81 N = 9 | 23–32 26.8 ± 1.94 N = 37 |

**Note:**
Data for *S.* cf. *barbertonensis* (= new species, see below) is based on all type and additional material. For ventral scale rows longitudinally, rare exceptions are indicated in parentheses, and superscripts indicate the relevant numbers of specimens. Femoral pores (males and females >95 mm SVL only) and differentiated femoral scales (generation glands, males >95 mm SVL only) are expressed as average number per thigh (left and right sides examined).

long (basal) tail and occipital spines, while all other species (including *S. warreni*, *S. barbertonensis* and *S.* cf. *barbertonensis*) had two-layer glands and relatively short spines. The latter two forms were found to be more spinose than *S. warreni* (i.e. longer occipital scales and proximal caudal spines).

### Statistical analyses

Both principal components and linear discriminant analyses reveal clear separation in scale characters between *S. warreni* and *S. barbertonensis*, and *S.* cf. *barbertonensis* and *S. barbertonensis* (4% LDA mis-classification rate in both cases) (Figs. 4C and 4D; Tables S3 and S4). *Smaug* cf. *barbertonensis* and *S. warreni* display similar pholidosis and head proportions and cannot be consistently sorted by these characters alone (25% LDA mis-classification rate). The first two principal components explain 32% of the variation in the dataset.

## SYSTEMATICS

Family Cordylidae Gray, 1838

***Smaug swazicus*** Bates & Stanley sp. nov.

Swazi Dragon Lizard

Figures 5-8, Tables 2 and 3

urn:lsid:zoobank.org:act:A942675E-5E76-4FC9-AA8F-BFA7A4C131C7

*Cordylus warreni barbertonensis* (not *Van Dam, 1921*): *FitzSimons, 1943*: 426 (part: Hluti–Goedgegun, Eswatini); *Branch, 1988*: 164 (part) & *1998*: 195 (part); *Jacobsen, 1989* (part: Godlwayo; Nzulase; Farm Zwartkloof 60 HU); *Adolphs, 1996*: 15 (part); *Bourquin, 2004*: 96 (KwaZulu–Natal); *Adolphs, 2006*: 22 (part).

*Smaug warreni barbertonensis* (not *Van Dam, 1921*): *Stanley et al., 2011*: 64 (part); *Bates et al., 2014*: 211 (part, including photo on p. 211); *Reissig, 2014*: 190 (part, including figs. 215–217, 219).

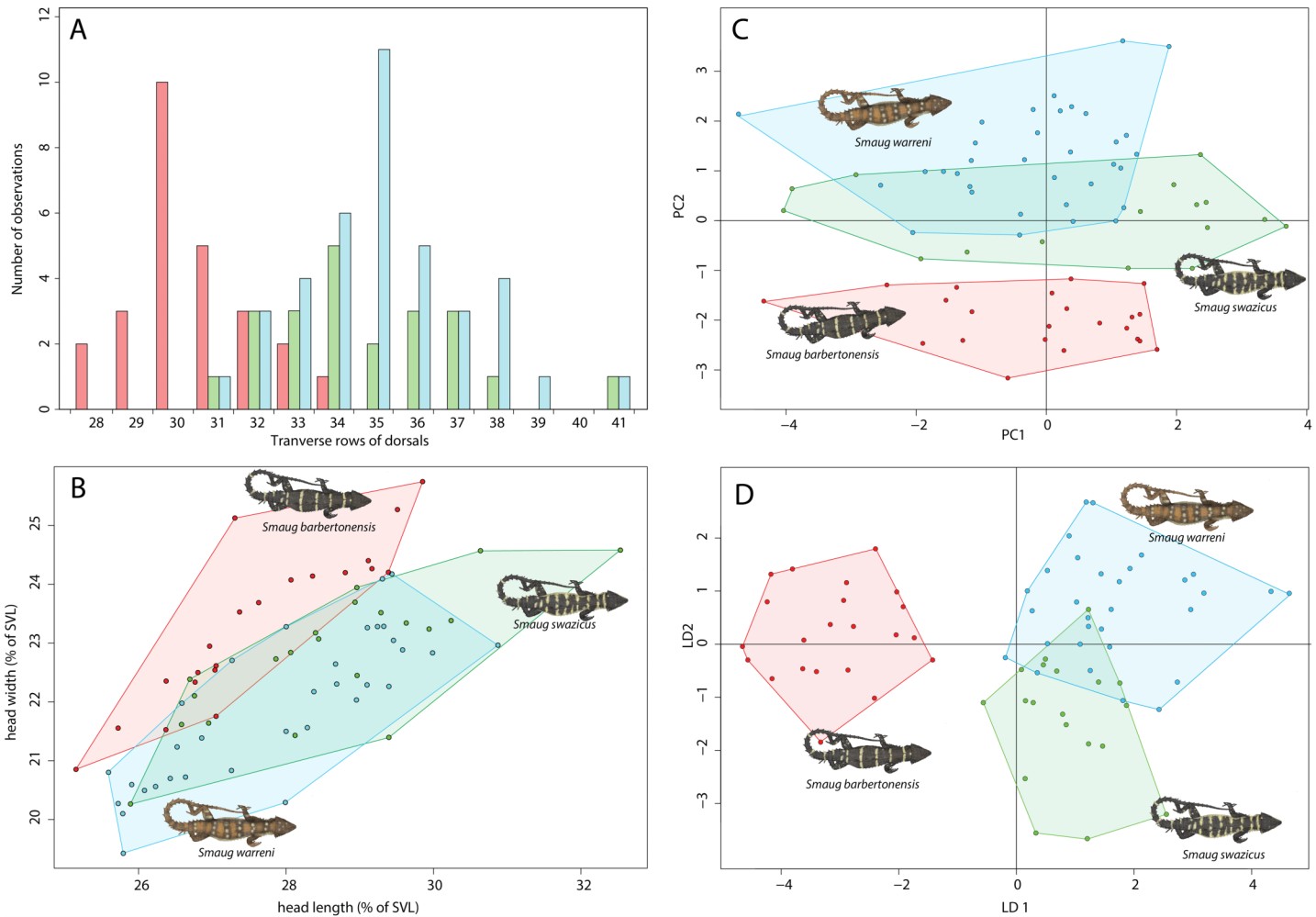

**Figure 4 Morphological variation in the *Smaug warreni* species complex.** (A) Bar graph showing numbers of dorsal scale rows transversely (*S. warreni*, N = 39; *S. barbertonensis*, N = 26; *S. swazicus* sp. nov., N = 22). (B) Scatterplot showing head length and head width (corrected by SVL) (>70 mm SVL: *S. warreni*, N = 33; *S. barbertonensis*, N = 21; *S. swazicus* sp. nov., N = 20. (C) Principal Component analysis of 13 meristic and three linear characters (*S. warreni*, N = 31; *S. barbertonensis*, N = 23; *S. swazicus* sp. nov., N = 18). (D) Linear Discriminate analysis of 13 meristic and three linear characters (*S. warreni*, N = 31; *S. barbertonensis*, N = 23; *S. swazicus* sp. nov., N = 18). In all graphs, *S. warreni* in blue, *S. barbertonensis* in red, *S. swazicus* sp. nov. in green. (Graphs produced by: E.L. Stanley).

*Smaug* sp. *Stanley & Bates, 2014*: 905.
*Smaug barbertonensis* (not *Van Dam, 1921*): *Mouton et al. 2018*: 460, fig. 2g.
*Smaug* cf. *barbertonenis Mouton et al., 2018*: 464.

**Holotype.** NMB R9201 (Figs. 5–7; sample from this specimen was used in molecular analysis by *Stanley & Bates, 2014*), adult male (differentiated glandular femoral scales present; mid-ventral incision present) from 320 m SSE of car park, Maguga Dam, Hhohho Region, Eswatini (26°04′57″S, 31°15′59″E; 2631AB; 635 m a.s.l.), collected by E.L. Stanley & J.M. da Silva, 31 October 2008.

**Paratypes.** Allotype: TM 78918 (Figs. 2B and 2E), adult female (no differentiated femoral scales) from Nkomati Gorge, Malolotja Nature Reserve, Hhohho Region,

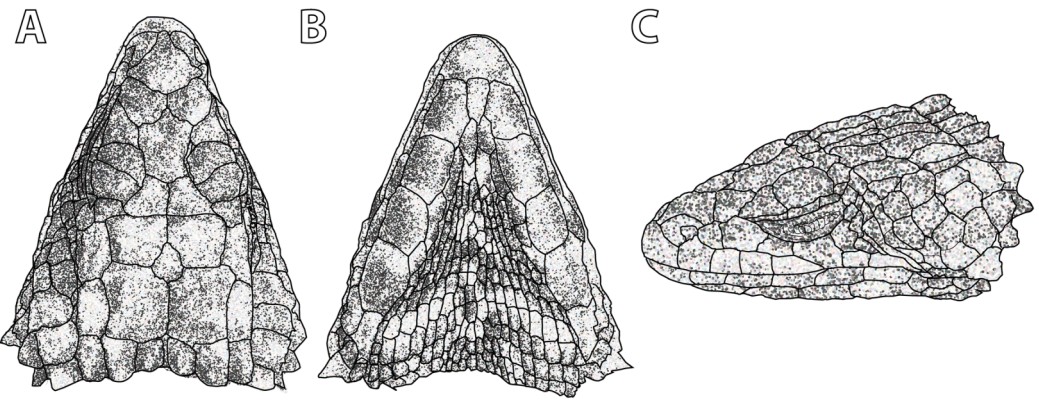

**Figure 5** *Smaug swazicus* **sp. nov. (A) Dorsal, (B) ventral and (C) lateral views of the head of the holotype (NMB R9201).** (Drawing credit: E.L. Stanley).

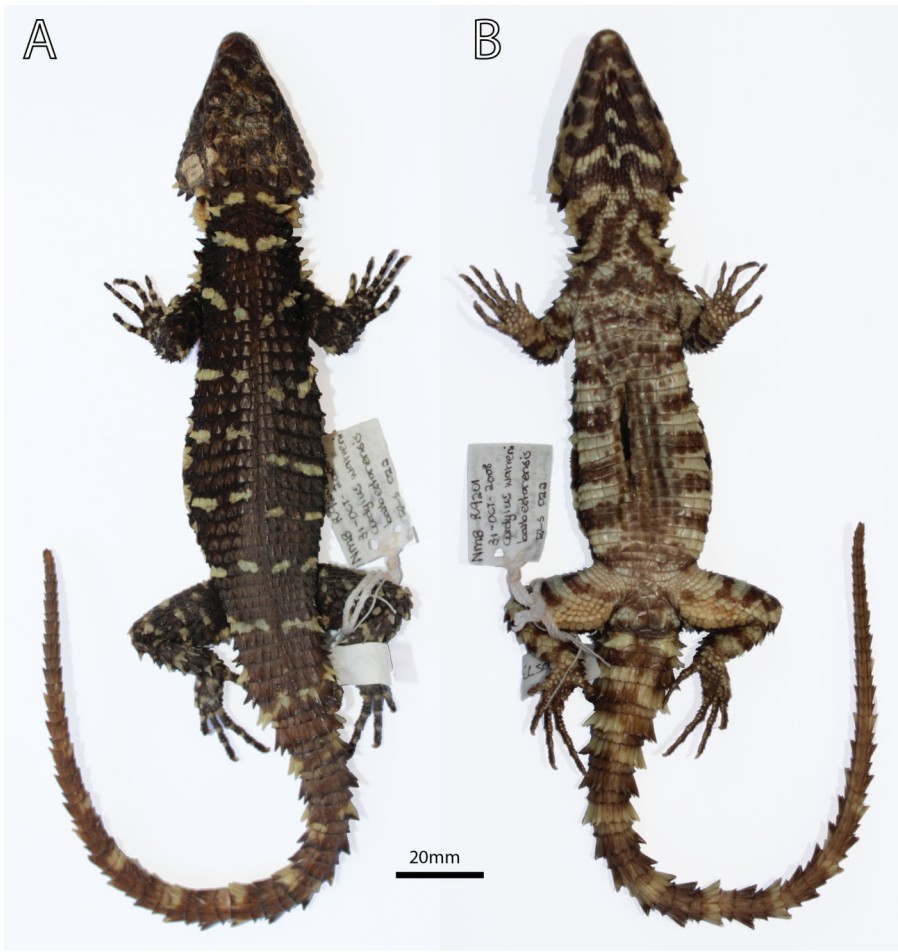

20mm

**Figure 6** *Smaug swazicus* **sp. nov. Holotype (NMB R9201, male). (A) Dorsal view. (B) Ventral view.** (Photo credits: M.F. Bates).

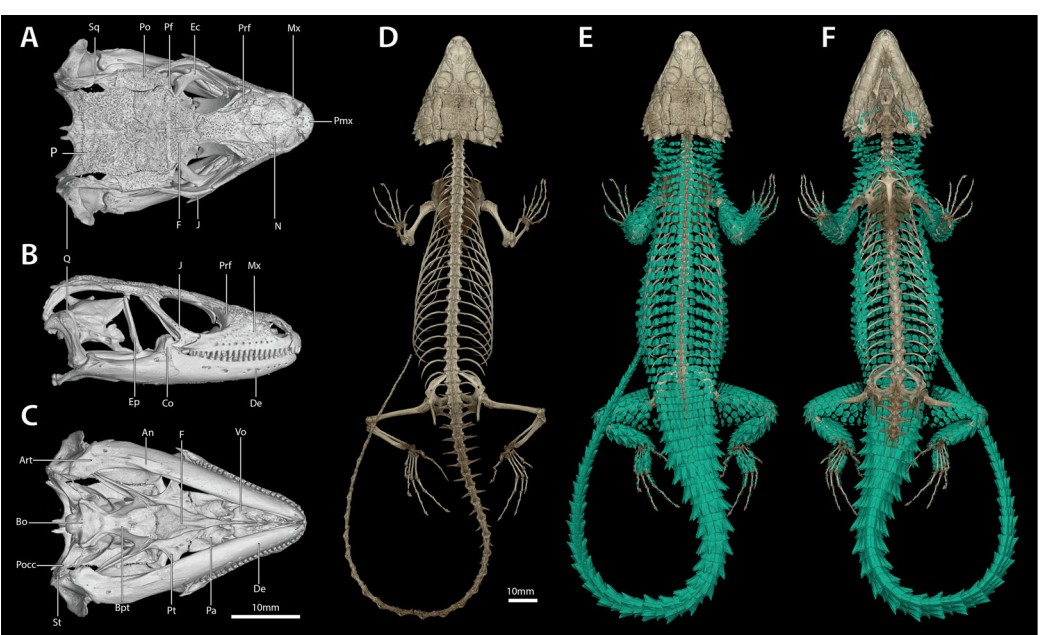

**Figure 7 Cranial (A–C), postcranial (D), and dermal (E, dorsal and F, ventral) osteology of *Smaug swazicus* sp. nov. (holotype, NMB R9201).** Body osteoderms are highlighted in blue-green. An, angular; Art, articular; Bo, basioccipital; Bpt, basipterygoid; Co, coranoid; De, dentary; Ec, ectopterygoid; Ep, epipterygoid; F, frontal; J, jugal; Mx, maxilla; N, nasal; P, parietal; Pa, palatine; Pf, postfrontal; Pmx, premaxilla; Po, postorbital; Pocc, paraoccipital; Prf, prefrontal; Pt, pterygoid; Q, quadrate; Sq, squamosal; St, supratemporal; Vo, vomer. (Images produced by: E.L. Stanley).

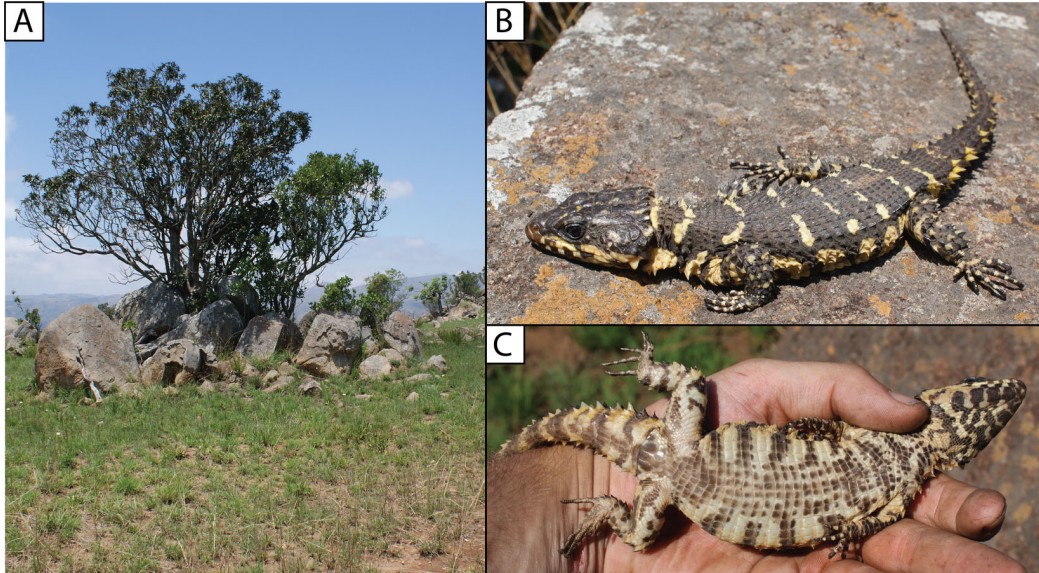

**Figure 8 (A) Shady rocky outcrops at Nkomati Viewpoint, Malolotja National Park, Eswatini, typical habitat of *Smaug swazicus* sp. nov. (B) Dorsal colouration of live paratype of *S. swazicus* (NMB R9194). (C) Ventral colouration of *S. swazicus*.** (Photo credits: E.L. Stanley).

Table 2 Mensural data (mm) for the type series of *Smaug swazicus* sp. nov. (M = male, F = female, J = juvenile; r = regenerating tail).

| Museum number | Type status | Sex | Snout-vent length | Tail length | Total length | Head length | Head width | Head depth |
|---|---|---|---|---|---|---|---|---|
| NMB R9201 | Holotype | M | 138.76 | 187 | 325.76 | 40.19 | 31.15 | 16.19 |
| TM 78918 | Allotype | F | 143.80 | 161 | 304.80 | 37.23 | 29.14 | 15.97 |
| TM 83000 | Paratype | M | 129.35 | 150 | 279.35 | 34.86 | 27.99 | 14.85 |
| TM 83532 | Paratype | F | 132.08 | | | 35.11 | 28.55 | 15.08 |
| TM 42531 | Paratype | F | 102.77 | | | 29.23 | 23.71 | 12.25 |
| TM 51376 | Paratype | M | 129.60 | | | 36.37 | 29.60 | 17.71 |
| TM 78931 | Paratype | J | 70.14 | | | 21.00 | 16.30 | 8.00 |
| TM 78921 | Paratype | J | 65.76 | | | 19.33 | 14.07 | 8.14 |
| NMB R9194 | Paratype | M | 139.95 | 202 | 341.95 | 40.53 | 33.51 | 17.92 |
| NMB R9195 | Paratype | M | 145.01 | | | 41.96 | 34.36 | 18.71 |
| NMB R9202 | Paratype | M | 141.23 | 140r | 281.23+ | 40.11 | 32.73 | 16.61 |
| TM 73290 | Paratype | F | 140.11 | 116.40r | 256.51+ | 37.49 | 30.97 | 17.86 |

Table 3 Meristic data for the type series of *Smaug swazicus* sp. nov. (H = holotype, A = allotype, P = paratype).

| Museum number | Type status | Dorsal scale rows longitudinally | Dorsal scale rows transversely | Ventral scale rows longitudinally | Ventral scale rows transversely | Lamellae under 4th toe | Femoral pores (left/right) | Differentiated femoral scales (left/right) | Gular scales (betwn poster. sublabials) |
|---|---|---|---|---|---|---|---|---|---|
| NMB R9201 | H | 20 | 34 | 14 | 26 | 16 | 10/11 | 21/25 | 23 |
| TM 78918 | A | 21 | 34 | 14 | 28 | 18 | 10/10 | 0/0 | 25 |
| TM 83000 | P | 22 | 34 | 14 | 23 | 16 | 10/10 | 10/9 | 28 |
| TM 83532 | P | 25 | 41 | 14 | 29 | 18 | 11/10 | 0/0 | 27 |
| TM 42531 | P | 22 | 37 | 14 | 25 | 17 | 10/10 | 0/0 | 28 |
| TM 51376 | P | 26 | 36 | 12 | 26 | 18 | ?/11 | ?/24 | 26 |
| TM 78931 | P | 21 | 35 | 14 | 26 | 17 | 12/12 | 10/16 | 25 |
| TM 78921 | P | 23 | 32 | 14 | 25 | 18 | 11/11 | 12/12 | 24 |
| NMB R9194 | P | 22 | 34 | 12 | 26 | 17 | 11/12 | 26/35 | 22 |
| NMB R9195 | P | 25 | 35 | 14 | 27 | 17 | 12/11 | 33/? | 25 |
| NMB R9202 | P | 22 | 34 | 14 | 27 | 18 | 12/11 | 15/19 | 24 |
| TM 73290 | P | 24 | 37 | 14 | 27 | 19 | 11/10 | 0/0 | ? |

**Note:**
Values on either side of a slash refer to the animal's left and right sides respectively.

Eswatini (26°03′15″S, 31°08′06″E; 2631AA; 640 m a.s.l.), collected by R.C. Boycott, 29 August 1993. Ten more paratypes: TM 83000, adult male, 1 km NW of Maguga Dam, Hhohho Region, Eswatini (26°04′04″S, 31°14′55″E; 2631AA; 618 m a.s.l.), R.C. Boycott, 25 March 1997; TM 83532, adult male, 5 km SE of Bhunya, Manzini Region, Eswatini (26°32′16″S, 31°02′54″E; 2631CA; 960 m a.s.l.), R.C. Boycott, 28 June 2000; TM 42531, adult female, Mbutini Hills, 23 km N of Sepofaneni, Manzini Region, Eswatini (26°31′34″S, 31°35′45″E; 2631DA), W.D. Haacke, 3 September 1972; TM 51376, adult male, 15 km NW of Gilgal on route to Manzini, Manzini Region, Eswatini (2631DA), W.D. Haacke,

3 September 1972; TM 78931, juvenile, Nkomati Gorge, Malolotja Nature Reserve, Hhohho Region, Eswatini (26°03′14″S, 31°08′02″E; 2631AA; 669 m a.s.l.), R.C. Boycott, 14 September 1993; TM 78921, juvenile, Nkomati Valley, Hhohho Region, Eswatini (26°03′12″S, 31°14′24″E; 2631AA; 580 m a.s.l.), J. Linden, 31 October 1992; NMB R9194, adult male, Nkomati Viewpoint, Malolotja Nature Reserve, Hhohho Region, Eswatini (26°04′29″S, 31°07′32″E; 2631AA; 1,033 m a.s.l.), E.L. Stanley & J.M. da Silva, 31 October 2008 (Fig. 8B); NMB R9195, adult male, Nkomati Viewpoint, Malolotja Nature Reserve, Hhohho Region, Eswatini (26°04′55″S, 31°08′03″E; 2631AA; 1,139 m a.s.l.), E.L. Stanley & J.M. da Silva, 31 October 2008; NMB R9202 (mid-ventral incision present; sample from this specimen was used in molecular analysis by *Stanley & Bates, 2014*), adult male from 230 m SSE of car park, Maguga Dam, Hhohho Region, Eswatini (26°04′54″S, 31°15′58″E; 2631AB; 640 m a.s.l.), collected by E.L. Stanley & J.M. da Silva, 31 October 2008; TM 73290, adult female, Nzulase, Mpumalanga Province, South Africa (25°51′S, 31°38′E; 2531DC), N.H.G. Jacobsen, 29 March 1983.

**Additional records** (*material examined). SOUTH AFRICA: KwaZulu–Natal, Province. Godlwayo Hill (27°20′S, 31°25′E; 750 m a.s.l.) TM 73290-1*, 73294*; Ithala Game Reserve (central point for mapping 27°30′S, 31°17′E) TM 51670*; Farm Zwartkloof 60HU (27°24′S, 31°33′E; ?420 m a.s.l.) TM 73285*. ESWATINI: between Hluti and Goedgegun (now called Nhlangano) in Shiselweni Region (no co-ordinates) TM 16827–9*, 16798*; same locality name as previous (27°12′20.1″S, 31°20′10.5″E, photographic record: T. Sparkes); 1 km NW of Maguga Dam wall (26°04′04″S, 31°14′55″E; 618 m a.s.l.) DNSM 1707 (identified as *Smaug barbertonensis* by R.C. Boycott, 2019, personal communication), TM 83002*; 1 km SE of Maguga Dam wall (26°05′08″S, 31°16′20″E) DNSM 1710 (identified as *Smaug barbertonensis* by R.C. Boycott, 2019, personal communication); Manzini, 25 km ESE of (26°31′S, 31°37′E) NMZB–UM 2026, 2529 (identified as *Cordylus warreni barbertonensis* by D.G. Broadley, 2014, personal communication—scalation details below); Nwempisi Gorge, 12 km E of Mankayane (26°42′13.4″S, 31°11′49.1″E, sight record: R.C. Boycott, 2019, personal communication). NO DATA: AMNH-R173382 (used for CT scanning).

**Diagnosis.** (includes 'additional material') Distinguished from all other cordylids (Cordylidae) by its unique combination of dorsal, lateral and ventral colour patterns (see descriptions and figures). Referable to the genus *Smaug* on the basis of its large size and robust body, enlarged and spinose dorsal and caudal scales, enlarged occipital scales, and frontonasal in contact with the rostral, separating the nasal scales.

  A medium to large species of *Smaug* distinguishable by the following combination of characters: (1) back dark brown usually with 5–6 pale bands (usually interrupted) between fore- and hindlimbs, each band consisting of pale, sometimes dark-edged, markings; (2) pale band on nape behind occipitals; (3) flanks with large pale spots or blotches; (4) belly pale with a dark median longitudinal band bordered on either side by broad, dark, bands; (5) throat pale with extensive bold brown mottling (sometimes forming transverse bands; often much of throat is dark); (6) six enlarged, moderately to

non-spinose, occipital scales, middle pair the smallest, outer occipitals usually shorter than the adjacent inner ones; (7) dorsolateral and lateral scales moderately spinose; (8) tail moderately spikey; (9) dorsal scale rows transversely 31–41; (10) dorsal scale rows longitudinally 20–26; (11) ventral scale rows transversely 23–29; (12) ventral scale rows longitudinally 14 (rarely 12); (13) femoral pores per thigh 10–13; subdigital lamellae on 4th toe 16–19.

Its status as a new species is also supported by monophyly with high levels of support from three mitochondrial and eight nuclear markers (see *Stanley & Bates, 2014*; using samples from NMB R9201–2).

It differs from the terrestrial *S. giganteus* by its smaller adult size (maximum SVL 145 mm vs. 198 mm), and possession of six moderate sized and weakly spinose occipitals, vs. four (occasionally five) large and distinctly spinose occipitals. Differs from other species of *Smaug* as follows: from *S. vandami* by having six (versus usually four) occipitals; from *S. depressus* by having only 10-13 (vs. 16-24) femoral pores per thigh in males; and from *S. breyeri* by having much less rugose head shields. It differs from *S. giganteus*, *S. breyeri* and *S. vandami* by having less spinose occipitals and tail spines, and two-layer (rather than multi-layer) generation glands. Differs from *S. mossambicus* and *S. regius* by having the first supralabial with moderate or no (vs. distinct) upward prolongation and lacking obvious sexual dichromatism (only males of the latter two species have bright yellow to orange flanks).

Most similar to *S. barbertonensis* and *S. warreni*, but easily distinguishable by its colour pattern (as described above) compared to *S. barbertonensis* (back dark brown with 4–5 pale bands between the limbs, pale spot or blotch on nape behind occipitals; flanks dark with narrow pale vertical markings; venter mostly dark brown or black) and *S. warreni* (back usually pale brown with 5–6 pale dark-edged bands between the limbs, pale band on nape behind occipitals; flanks pale with brown markings; venter with brown markings on most scales) (Figs. 2, 6 and 8, and others below); by usually having short, blunt, non-spinose scales at the edges of the ear openings (usually elongate and spinose in *S. barbertonensis*); and quadrates with a pronounced ridge and concave region at the lateral edge of the *adductor musculus mandibulae* posterior origin (no pronounced ridge or concave region in the other two species). Also differs as follows: outer occipitals usually shorter than the adjacent inner ones (of about equal length in *S. warreni*); head narrower than *S. barbertonensis* (head width/head length = 76–84% vs. 80–92% in adults); generally higher numbers of transverse dorsal scale rows (32-37 in 86% of specimens) than *S. barbertonensis* (29-32 in 81% of *S. barbertonensis*).

**Description of holotype.** NMB R9201. *External morphology:* Snout-vent length 138.8 mm, tail length (original) 187 mm, total length 325.8 mm, head length 40.2 mm, head width 31.2 mm, head depth 16.2 mm. Tail length/SVL = 135%; head width/SVL = 22.4%; head width/head length = 77.5%; head depth/head length = 40.3%. Head strongly depressed, head shields rugose and moderately striated over parietal region. Frontonasal 1.05 times as wide as long, in contact with the rostral and loreals, separating the nasals, latter slightly swollen. Nostril—with a large inner flap attached posteriorly—situated

in the posterior part of the nasal and in contact with the loreal and 1st supralabial. Prefrontals in contact at their inner angles, separating the frontal from the frontonasal. Frontal hexagonal, slightly widened anteriorly, anterior sides curved slightly inwards. Frontoparietals slightly broader than long. Posterior parietals larger than anterior ones; interparietal between the four parietals, more sharply pointed anteriorly than posteriorly. Occipital scales 6, well-developed, bluntly spinose, the outer ones shorter and smaller than the adjacent inner ones, middle pair shortest and narrowest. Anteriormost of three upper temporals large, keeled at its lower edge. Gulars 23. Lateral temporals large, often bluntly keeled. Scales at anterior edge of ear opening (four on left side of head, five right) projecting outwards as flattened and somewhat spatulate spines, the lowermost one narrow and somewhat slender, the one above it distinctly spatulate and the largest; the other scales shorter and somewhat blunt. Supraoculars 4, the anterior one longest, the next (2nd) one broadest, posterior one the smallest. Supraciliaries 4, anterior one the longest, posterior shortest. Lower eyelid opaque, consisting of about 10 small, vertically-elongated, scales. Preocular at least twice the size of the loreal. Four large scales below the eye. Rostral 2.04 times wider than deep. Supralabials 6, 4th (longest) and 5th separated by a large suborbital shield (much narrower below than above). Mental 1.36 times as wide as long. Infralabials 6 (5th and 6th somewhat keeled), bordered below by five large sublabial shields. First pair of sublabials separated by an elongated scale in contact with the large mental and followed behind by two pairs of similar-sized scales and numerous small elongated scales that increase in size until about the middle of the throat, but then reduce in size posteriorly. Sides of neck with irregular erect spines, the largest about twice as high as wide. Dorsal scales large, rugose, often striated, forming regular (but not always aligned) transverse series; four vertebral rows with smooth scales (probably due to rubbing against rocks), other dorsals keeled, but dorsolateral and lateral scales keeled and moderately spinose. Dorsals in 34 transverse series (from first row posterior to occipitals to row above vent) and 20 longitudinal rows. Ventrals smooth, mostly quadrangular, occasionally pentagonal near middle of venter posteriorly, mostly broader than long, two outer rows moderately keeled and weakly spinose, some scales of the 3rd row also weakly keeled, forming 14 longitudinal and 26 transverse series from axil to groin (with an additional seven rows to base of throat). A pair of enlarged hexagonal preanal plates (slightly longer than wide), with smaller plates anteriorly and on the sides. Limbs above with large, keeled, strongly spinose scales. Femoral pores 21 (10 left leg, 11 right). Differentiated femoral scales 46 (21 left, 25 right). Fourth toe on each foot with 16 subdigital lamellae. Tail with whorls of large, strongly keeled, spinose scales; each whorl separated by a smaller whorl of small, moderately keeled and weakly spinose scales; two upper lateral caudal scale rows consist of especially large and very strongly spinose scales (spines project backwards at angles of about 45°); subcaudal scales of the larger whorl elongate and narrow, those of the smaller whorl much shorter, mostly pentagonal (occasionally rectangular), moderately keeled, those in the centre only weakly spinose.

*Colour*: (similar in life and in preservative; Fig. 6) Back dark brown to black with cream to yellow markings forming five interrupted transverse bands (with only slightly dark

borders in preservative) between fore-and hindlimbs, which continue along the tail, together with an incomplete pale band immediately behind the occipitals and another on the nape that is divided medially and curved slightly backwards. Belly cream-white with a brown longitudinal band medially (six ventral plates wide) and short, broad, widely separated brown bands on either side between the limbs (at least five on the left, four on the right) which are often confluent with the darker parts of the back. The joining of these dark ventral and dorsal markings decorates the flanks with large cream-yellow spots/ blotches. Top of limbs dark brown with numerous irregular cream to yellowish spots and blotches; underparts of limbs mostly cream-white with irregular brown markings, occasionally bands. Top of head brown with scattered irregular cream markings; throat mottled in dark brown and cream-white, most dark markings forming four irregular, wavy transverse bands.

*Cranial skeleton*: (Fig. 7) Scales of the dorsal and temporal regions of the skull and the ventrolateral aspects of the jaws are underlain with rugose osteoderms. These osteoderms fuse to the proximal parietal, frontal and postorbital bones, although the mesokinetic and metakinetic joints appear unobstructed and flexible. Lateral maxilla and anterior aspect of the premaxilla lack osteoderms. The parietal is pentagonal, with five osteoderms that underlie the parietal shields fused to its dorsal surface and a bifid medioposterior process that extends either side of the sagittal crest of the supraocciptial. Three large osteoderms are fused to the frontal, which is unpaired and clasped by the parietal at its posterolateral edge. The upper temporal fenestra is obscured anteriorly by a large osteoderm fused to the dorsal surface of the postorbital bone, and posteriorly by an unfused rectagonal osteoderm that overlies the squamosal. Premaxilla is unpaired and contains seven pleurodont teeth and five foramina, with a dorsal process that extends posteriorly to be clasped by the nasals, which themselves insert into the frontal. The maxilla is scinciform, with a deeply grooved crista dentalis, nine left or eight right lateral foramina, and 19 teeth. Teeth display pleurodont attachment and are unicuspid, with a slight concave surface where they connect with the mandibular teeth. No palpebral is present and the prefrontal connects directly to the anteriormost superorbital osteoderm. The jugal is triangular in cross-section and asymmetrically T-shaped, with a tapering anterior process and a broad, truncated posterior process that extends along and past the posterior edge of the maxilla. Lacrimal bone is small, flattened and oval. Pterygoids are edentate and extend back to connect with the quadrates, becoming C-shaped in cross-section posterior to the epipterygoid condyle. The squamosal is curved and blade-like, circular in cross-section anteriorly, becoming flattened posteriorly, where it articulates with the cephalic condyle of the quadrate and the braincase. Supratemporals are flattened, ovoid and not fused with the elongate paroccipital processes. The posterior aspect of the prootic is not fully fused with the oto-occipital, resulting in a deep groove along the dorsal aspect of the para-occipital processes. Quadrates are very broad with a pronounced ridge and concave region at the lateral edge of the *adductor musculus mandibulae* posterior origin. The supraoccipital has a strong sagittal crest that extends posteriorly to contact the ventral surface of the medioposterior process of the parietal.

The prootic bears an extended alar process and a well-developed, rhomboid christa prootica, and a very weak supratrigeminal process. Basipterygoid processes are well developed and flattened. The lower jaw possesses a large adductor fossa, a highly flattened and medially extended retroarticular process, a medially open Meckelian canal that is closed posteriorly by a large splenial and a dentary with a strong subdental shelf; 21 mandibular teeth, and nine dentary foramina.

*Postcranial skeleton*: (Fig. 7) Tail complete, 26 presacral vertebrae, 32 caudal vertebrae. The haemapophyses of the first caudal osteoderms extend laterally to fuse to the posteroventral edge of the parapothysis, forming a biphid rib. Four cervical, three sternal, two xiphisternal, six left and seven right long asternal ribs with ossified costal cartilage, then six left and five right short asternal ribs and one very short pair of ribs immediately anterior to the sacral vertebrae. Cervical ribs 2-4 are distally flattened and biphid, with the ventral processes more elongated. Pubis flattened and curved with a large, ventrally angled pectineal tubercle. Pubic symphysis flattened and triangular, separating the pubes entirely. Hyperischiam and hypoischium well developed. Illium triangular in cross-section, with a feeble iliac tubercle. Sternal plate broad with no fontanelle. Interclavicle cruciform, clavicles flattened dorsally. Epicoracoid connects the scapular ray to the primary and secondary coracoid rays, but not to the anterior process of the scapular. Phalanges display a typical pattern of 2-3-4-5-3 for the manus and 2-3-3-5-4 for the pes. Metatarsal 5 with elongated medial process at midbody.

*Dermal osteology*: (Fig. 7) Dorsal and lateral sides of the trunk are covered in circular, well-separated osteoderms, dorsomedially unkeeled grading to well keeled and mucronate towards the sides. The nuchal osteoderms are small, becoming highly spined posterior to the tympanic opening. Ventral osteoderms are delicate and plate-like and restricted to the gular and anterior pectoral regions. The forelimbs are covered in keeled, non-imbricate, circular/rhomboid osteoderms, while the hindlimbs are well armoured, except for the ventral surface of the thigh which lacks osteoderms. Osteoderms on the posterior part of the hindlimbs are heavily spinose. The caudal osteoderms are large, robust and arranged in imbricated whorls. Caudal osteoderms are feebly keeled and mucronate along the dorsal and ventral aspects, becoming more heavily spined laterally.

**Variation in paratypes** (including allotype TM 78918; Figs. 2B and 2E). *External morphology:* Tail length/SVL 112-144% (SVL: 129.4-143.8 mm, N = 3); head width/ SVL = 21.6-23.9% in males (SVL: 129.4-145.0 mm, N = 5), 20.3-23.1% in females (SVL: 102.8-143.8 mm, N = 4); head width/head length = 78.3-82.7% (SVL: 102.8-145.0 mm, N = 9); head depth/head length = 41.9-48.7% (SVL: 102.8-145.0 mm, N = 9) (Tables 2 and 3). In TM 78918, shields on anterior part of head smooth, weakly rugose on posterior part of head but without striations; in two juveniles: head shields smooth (TM 78921) or weakly rugose without striations (TM 78931). Frontonasal 0.89-1.12 (0.94-1.05 in juveniles) times as wide as long. Nasal scale fragmented on left side in TM 83000. Small infranasal present on both sides of head in TM 78918. Frontal with anterior sides straight in TM 78921, strongly curved inwards in NMB R9202, separated from

rostral by a small rectangular scale in TM 42531. Prefrontals in narrow contact in TM 51376, anterior half of prefrontals in contact in TM 78918 and 83532. Frontoparietals about as wide as long in TM 73290, 78918, 78931 and 83532. Interparietal sunken in NMB R9195, about as large as an anterior parietal in two juveniles (TM 78921, 78931), triangular in TM 51376 and 73290, and as pointed posteriorly as it is anteriorly in TM 83000 and 83532. Occipitals 6 but middle pair separated by a small elongate scale in TM 78931, mostly very weakly spinose, all of about the same size in TM 78918, outer scale and the one adjacent to it of similar length in NMB R9202 and TM 73290; in TM 83532 scales of the middle pair are shortest but of similar width to the others, and wider than the outer occipitals; but the middle and outer scales may be similar in size (TM 42531, 51376); middle occipitals the same size as second occipitals on either side in TM 78931; on left side of TM 83000 the outer occipital is about equal in size to the occipital adjacent to it; in TM 51376 the inner occipitals are rugose only, not spinose. Gulars 22-28 (25 in allotype). Posterior upper temporal scale keeled at its lower edge in TM 42531, 78918 and 78921; anterior and posterior upper temporals similar in size and keeled at the sides in TM 51376. Lateral temporals rugose only (not keeled) in NMB R9194 and TM 51376. Scales at anterior edge of ear opening 4–6 (3rd from the top is tiny in TM 83532), lowermost spine often not slender and similar to other small spines, but elongate and distinctly spiny in TM 42531. First and 2nd (TM 78921, 78931) and 1st and 3rd (TM 78918) supraoculars about the same length, 2nd and 3rd on left side of NMB R9194 largely fused. Supraciliaries 5 (left side: NMB R9202, TM 42531; right: TM 83532), first and second supraciliaries about equal in length in TM 78931. Lower eyelid transparent in TM 73290 and TM 83532, usually consisting of several irregular scales (e.g. NMB R9195). Preocular about 1.5 times (TM 42531, 51376, 83532) and 1.75 times (TM 73290) larger than loreal. Five large scales below the eye in TM 42531, 78921, NMB R9194-5, TM 73290 (left) and 83532 (left); large suborbital shield divided vertically in TM 73290 (left side). Rostral 2.14-2.81 (1.87-2.10 in juveniles) times wider than deep. Supralabials 7 on left side of head in TM 73290 and TM 78921; sixth (of six) in TM 51376 is granular and 2nd is fragmentary; 4th (of 6) distinctly keeled in TM 42531; 3rd and 4th fused in NMB R9202. Mental 1.17-1.66 (1.16 in juvenile TM 78931) times as wide as long. Fourth and 6th infralabial weakly keeled in TM 78931. Fifth and most posterior sublabial on either side of head rugose and keeled in TM 73290 and 78918; 1st pair of sublabials in contact (NMB R9202; TM 78918, 78921, 78931, 83000, 83532), or separated by a narrow groove (TM 51376), a large rectangular scale (NMB R9194), an elongated triangular scale (NMB R9195), or separated posteriorly by a tiny pair of granules (TM 73290); 1st pair of sublabials followed by three (not two) pairs of smaller, slightly enlarged scales in TM 78931, and by one pair of distinctly enlarged scales in NMB R9202. Spines on sides of neck only about 1.5 times (not twice) as high as wide in juveniles (TM 78921, 78931) and TM 83000. Dorsal scales of TM 78918 and 78931 with short folds rather than distinct striations; two vertebral scale rows smooth in TM 42531, 4-6 rows smooth in TM 83532, 6-8 rows smooth in NMB R9195, none smooth in TM 83000, all vertebrals keeled in juveniles. Dorsolateral and lateral scales usually keeled and moderately spinose, but weakly spinose in juveniles. Dorsals in 32-41 (34 in allotype) transverse, and 21-26 (21 in allotype) longitudinal, rows.

Ventrals occasionally pentagonal (TM 73290, 78918, 83532), longer than broad on anterior part of belly (TM 73290, 78918) or mostly square (NMB R9195, TM 51376 and 83000). All ventrals smooth in TM 51376; in NMB R9194 and TM 83532 only the outermost row of ventrals is moderately keeled and weakly spinose, with rows 2–3 very weakly keeled only; some scales of the 3rd row also very weakly spinose in NMB R9194 (including first inner row) and TM 73290, 78918, 83000; all three outer rows weakly keeled in NMB R9195. Ventrals in 23–29 (28 in allotype) transverse rows (6–9 additional rows on throat), and occasionally only 12 (NMB R9194, TM 51376) longitudinal rows. Enlarged hexagonal preanal plates 3 (TM 78918) or 4 (TM 83532); median preanal plates (pair) pentagonal in TM 42531, 51376, 78931 and 83000, heptagonal in TM 73290 (left side) which also has two extranumerary plates posterior to the large pair, and irregular in TM 78921; no enlarged plates anterior to median and lateral plates in TM 78931; enlarged median pair of plates in TM 42531, 51376, 78931 and 83000 much elongated, each plate about twice as long as wide. Femoral pores 20–24 (10–12 on each thigh, 10 in allotype), appearing as small, shallow pits in females. Differentiated glandular femoral scales in males 19–61 (9–35 per thigh). Fourth toe with 16–19 (18 in allotype) subdigital lamellae.

*Colour*: Dorsum dark brown to black in preservative. Back with 4–6 (usually 5) interrupted transverse bands between fore- and hindlimbs, which are without dark borders or with only feeble indications thereof after preservation in alcohol. In TM 73290 there is a squarish cream marking on the nape between the pale band behind the occipitals and the band on the neck. Belly with 5–6 brown crossbands on either side of the median band (comprised of at least six longitudinal rows of ventrals, sometimes eight at places (e.g. TM 51376)) which is prominent in the centre of the belly. Throat with bold, dark mottling or reticulations; occasionally some markings form transverse bands, and sometimes most of the throat is black, especially anteriorly (e.g. NMB R9195 and TM 83532).

**Variation in additional material.** (material examined only for the characters listed below). *External morphology* ($N$ = 10 unless otherwise indicated): Tail length/SVL 1.35–1.42 (SVL: 97.9–132.7 mm, $N$ = 3); head width/SVL = 23.3–24.6% in males (SVL: 97.9–130.7 mm, $N$ = 3), 21.4–22.7% in females (SVL: 123.9–132.7 mm, $N$ = 3); head width/head length = 76.2–83.9% (SVL: 97.9–132.7 mm, $N$ = 6); head depth/head length = 36.2–47.6% (SVL: 97.9–132.7 mm, $N$ = 6). Preoculars 1; supraoculars 4 ($N$ = 9); supraciliaries 4 (3 on right side in TM 73294; $N$ = 8); loreals 1 ($N$ = 9); suboculars 4–5 (4 left, 5 right in TM 83002 and 16798); supralabials (anterior to median subocular) usually 4 (5 in TM 16828; $N$ = 9); infralabials 6 ($N$ = 9); sublabials 5 (4 on left side in TM 16827; $N$ = 9); occipitals 6 (additional small median scale in TM 16798 and 83002); gulars 24–29 ($N$ = 9); frontal and frontonasal in broad contact in TM 16798; scales at anterior edges of ear openings elongate and distinctly spinose (rather than short and blunt) in TM 16828; dorsal scale rows transversely 31–38, longitudinally 21–24; ventral scale rows transversely 25–27 ($N$ = 9), longitudinally usually 14, but 12 in TM 73285 and 773291; femoral pores per thigh 11–13 (males, $N$ = 4), 10–13 (females, $N$ = 6); differentiated femoral scales (generation glands) in males 19–29 per thigh ($N$ = 3); lamellae under fourth toe 16–19.

For the two Eswatini specimens (NMZB–UM 2026, 2529) examined by D.G. Broadley: occipitals 6; dorsal scale rows transversely 36 and 38 respectively, longitudinally 22; femoral pores/thigh 10.

*Colour*: Similar to holotype. Back with 5-6 (4 in TM 73285) interrupted transverse bands (sometimes with slightly dark borders in preservative). Belly with 5-6 brown crossbands on either side of the median band (prominent in the centre of the belly). Throat with bold, dark mottling or reticulations; occasionally some markings form transverse bands.

**Size.** Largest male (NMB R9194 (paratype), Nkomati Viewpoint, Eswatini) 140.0 + 202 = 342 mm, but NMB R9195 (paratype, Nkomati Viewpoint) has SVL of 145.0 mm (tail broken/missing). Largest female (TM 78918 (allotype), Nkomati Gorge, Eswatini) 143.8 + 161 = 305 mm.

**Etymology.** Named for the Kingdom of Eswatini, the country where most of the species' range is located. Both 'eSwatini' and 'Swaziland' derive from the word *iSwazi*, after the name of an early chief, *Mswati II* (c. 1820–1868).

**Distribution.** Highveld and Middleveld of Eswatini in Hhohho, Manzini and Shiselweni Regions, and adjacent areas in the South African provinces of (eastern) Mpumalanga (in Nkomazi municipality) and (northern) KwaZulu–Natal (in uPhongolo and Abaqulusi municipalies) (Fig. 9) at elevations of 462 to 1,139 m a.s.l.

**Natural history.** Diurnal and rupicolous, living in deep, horizontal (or gently sloping) crevices in granitic rock along hillsides, usually in the partial shade of trees (Fig. 8A; see also *Jacobsen, 1989*). According to R.C. Boycott (in litt., 2019), rocky terrain in closed canopy bushveld is the preferred habitat in Eswatini. A specimen in Ithala Game Reserve in KwaZulu–Natal was photographed on a tree trunk (ReptileMAP, VM no. 152451). When grasped by the hind limb, an individual from the type series performed an unusual anti-predator behaviour by repeatedly flexing and extending the inhibited limb caudally, so as to pull the captors' digits directly onto the very sharp whorl of spines at the base of the tail (E.L. Stanley, 2008, personal observation).

**Note.** The photograph of a specimen of 'Smaug warreni barbertonensis' from 'Barberton' in *Bates et al. (2014)* is the same one used for Fig. 8B in the current paper (i.e. NMB R9194, paratype of *S. swazicus* sp. nov.).

**Smaug barbertonensis** (*Van Dam, 1921*)
Barberton Dragon Lizard
Figures 10–12, Figs. S1–S2

*Zonurus barbertonensis Van Dam, 1921*: 240 (Barberton) Holotype: TM 4273 (Figs. S1 and S2); *Power, 1930*: 14 & 17 (Barberton).
*Zonurus barbertonensis barbertonensis* FitzSimons, 1933 (by implication after describing *Zonurus barbertonensis depressus*).

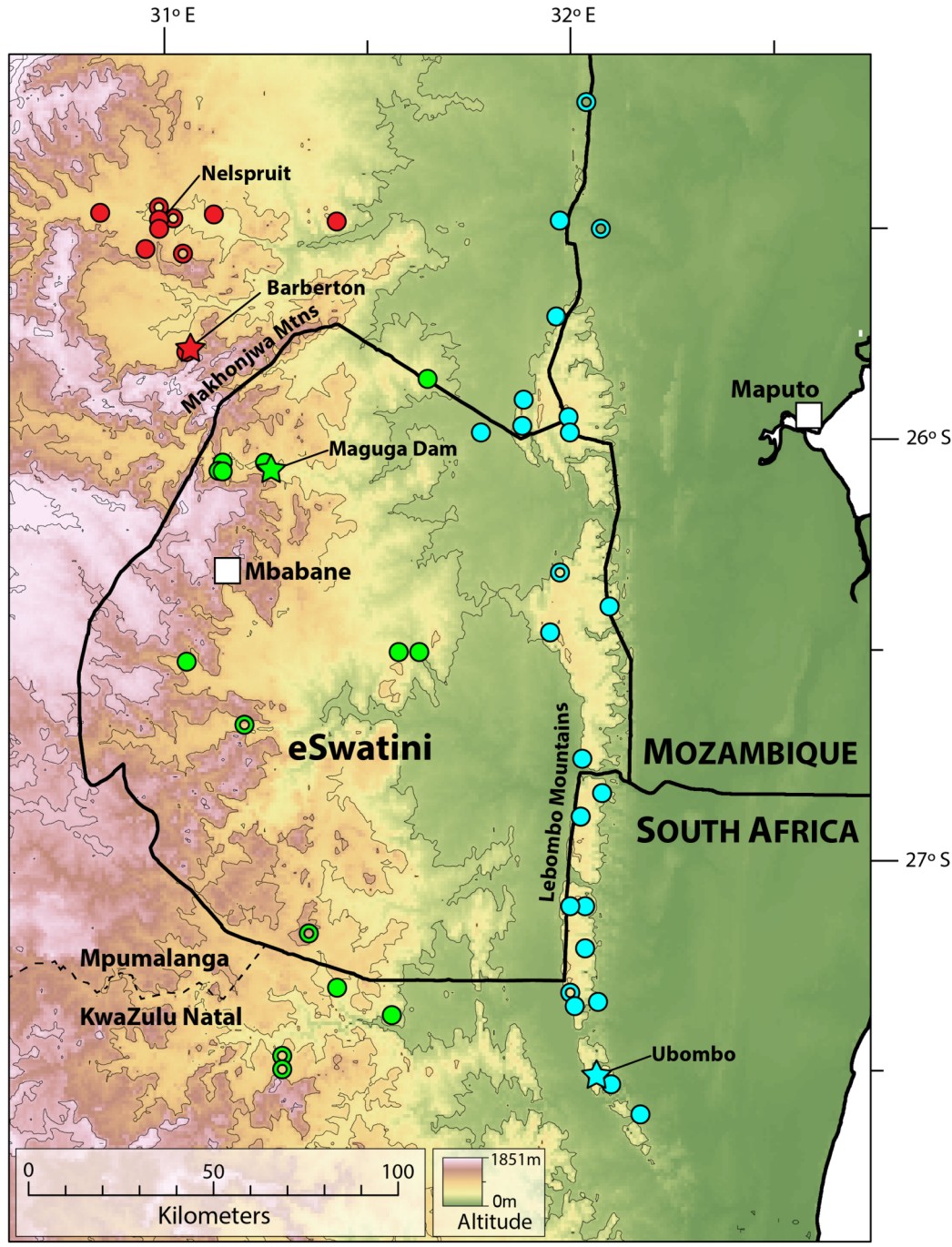

**Figure 9 Map showing localities for *Smaug barbertonensis* (red), *S. swazicus* sp. nov. (green) and *S. warreni* (blue).** Filled circles denote museum records, while open circles show geo-referenced photo vouchers from University of Cape Town's Animal Demography Unit Virtual Museum (Reptile-MAP), and other (mostly photographic) records (see text). The type locality for each species is represented by a star. Map created by: E.L. Stanley using ArcGIS® software by Esri, altitude layers from worldclim.org and Administrative boundaries from www.naturalearthdata.com.

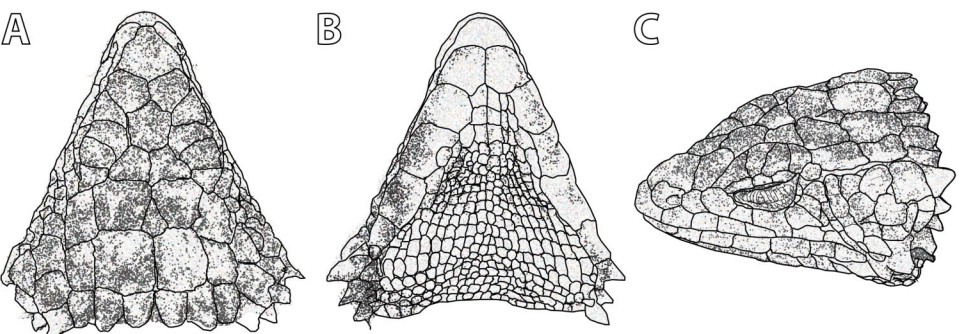

**Figure 10 *Smaug barbertonensis*. (A) Dorsal, (B) ventral and (C) lateral views of the head of NMB R9191 (topotype).** (Drawing credit: E.L. Stanley).

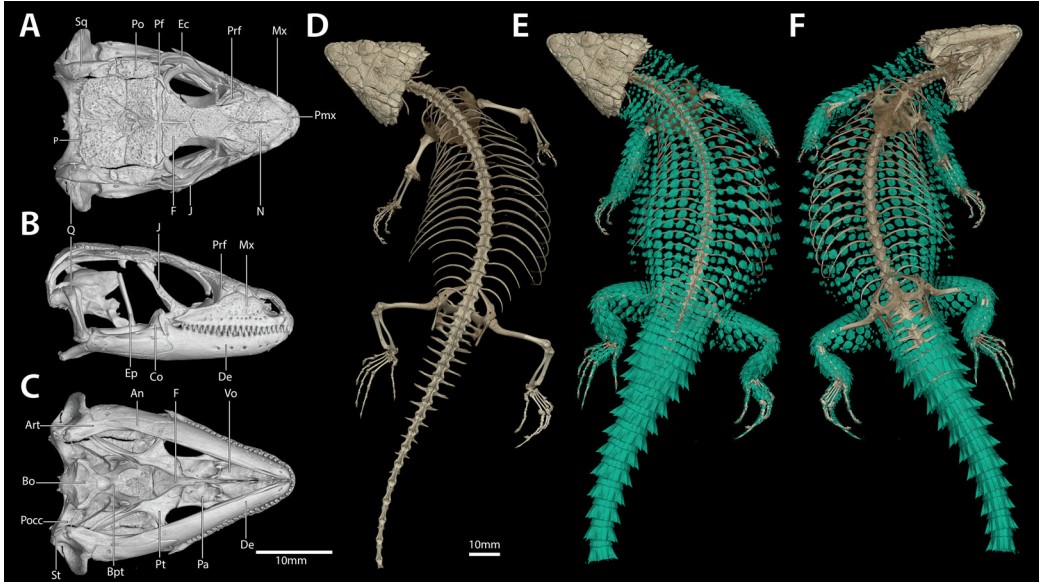

**Figure 11 Cranial (A–C), postcranial (D), and dermal (E, dorsal and F, ventral) osteology of *Smaug barbertonensis* (NMB R9196, topotype).** Body ostoderms are highlighted in blue-green. An, angular; Art, articular; Bo, basioccipital; Bpt, basipterygoid; Co, coranoid; De, dentary; Ec, ectopterygoid; Ep, epipterygoid; F, frontal; J, jugal; Mx, maxilla; N, nasal; P, parietal; Pa, palatine; Pf, postfrontal; Pmx, premaxilla; Po, postorbital; Pocc, paraoccipital; Prf, prefrontal; Pt, pterygoid; Q, quadrate; Sq, squamosal; St, supratemporal; Vo, vomer. (Images produced by: E.L. Stanley).

*Cordylus warreni barbertonensis FitzSimons, 1943*: 426 (part, Barberton and Nelspruit); *Loveridge, 1944*: 20 (Barberton); *Branch, 1988*: 164 (part), *1998*: 195 (part); *Jacobsen, 1989*: 590 (part: 5 km S of Nelspruit; Barberton Townlands 369JU; Broedershoek 129JU; Friedenheim 282JT; Karino to White River; Khandizwe; Nelspruit); *Adolphs, 1996*: 15 (part) & *2006*: 22 (part).

*Smaug warreni barbertonensis Stanley et al., 2011*: 64 (part); *Bates et al., 2014*: 211 (part, but excluding photo on p. 211); *Reissig, 2014*: 190 (part, including fig. 218).

*Smaug barbertonensis Stanley & Bates, 2014*: 905; *Mouton et al., 2018*: 463.

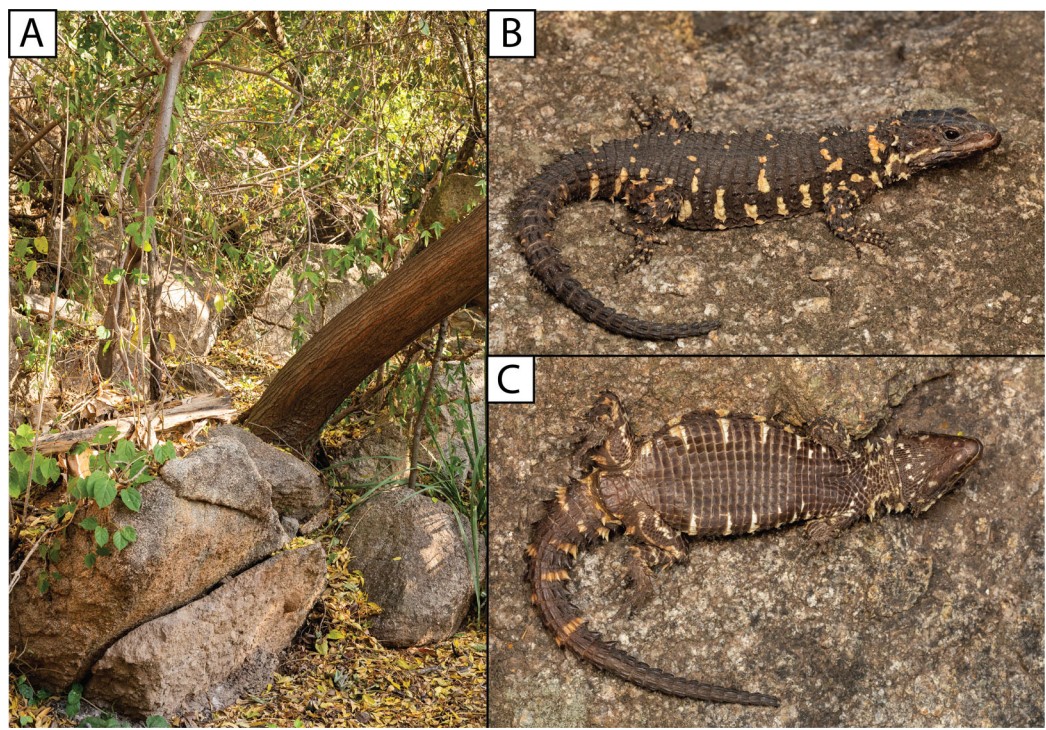

**Figure 12** (A) *Smaug barbertonensis* habitat: Nelspruit (Extension 5), Mpumalanga Province, South Africa. (B) Dorsal colouration of *S. barbertonensis* from the latter locality. (C) Ventral colouration of *S. barbertonensis* (same specimen). (Photo credits: Theo Busschau).

**Diagnosis.** Distinguished from all other cordylids by its unique combination of dorsal, lateral and ventral colour patterns (see descriptions and figures).

A medium to large species of *Smaug* distinguishable by the following combination of characters: (1) back dark brown to black with 4–5 bands (usually interrupted) between fore- and hindlimbs, each band consisting of pale, sometimes dark-edged, markings; (2) pale spot or blotch on nape behind occipitals; (3) flanks dark with narrow, pale, vertical markings; (4) belly mostly dark brown or black; (5) throat almost entirely dark brown or black with only a few pale areas; (6) usually six enlarged, moderately to non-spinose, occipital scales, middle pair the smallest, outer occipitals usually shorter than the adjacent inner ones; (7) dorsolateral and lateral scales moderately spinose; (8) tail moderately spikey; (9) dorsal scale rows transversely 28–34; (10) dorsal scale rows longitudinally 20–24; (11) ventral scale rows transversely 25–28; (12) ventral scale rows longitudinally 14 (rarely 16); (13) femoral pores per thigh 8–12; subdigital lamellae on 4th toe 15–19.

It differs from other species of *Smaug* as described above in the diagnosis of *S. swazicus* sp. nov. (but maximum SVL in *S. barbertonensis* is 140 mm, and femoral pores in males are 8–11 per thigh).

Most similar to *S. swazicus* sp. nov. and *S. warreni*, but easily distinguishable by its colour pattern (see comparisons in diagnosis of *S. swazicus* sp. nov. above); by usually having more elongate and spinose scales at the edges of the ear openings (shorter and

non-spinose in *S. swazicus* sp. nov. and *S. warreni*); and quadrates lacking a pronounced ridge and concave region at the lateral edge of the *adductor musculus mandibulae* posterior origin (with a pronounced ridge and concave region in *S. swazicus* sp. nov.). Also differs as follows: outer occipitals shorter than the adjacent inner ones (of about equal length in *S. warreni*); head relatively wider than the other two species (head width/head length = 80–92% vs. 73–84% in adults); lower numbers of transverse dorsal scale rows (29-32 in 81%) compared to *S. swazicus* sp. nov. (32-37 in 86%) and *S. warreni* (32-38 in 92%); and lower numbers of longitudinal dorsal scale rows (20-24, mean 21.7) compared to *S. warreni* (22-28, mean 23.6).

**Variation.** (*N* = 26 unless otherwise indicated) *External morphology*: (Fig. 10) Tail length/SVL 121-142% (SVL: 109.6-134.0 mm, *N* = 7); head width/SVL = 22.5-25.7% in males (SVL: 111.0-134.0 mm, *N* = 9), 20.9-24.1% in females (SVL: 109.6-139.9 mm, *N* = 11); head width/head length = 80.4-92.0% (SVL: 109.6-139.9 mm, *N* = 20); head depth/head length = 38.2-48.2% (SVL: 109.6-139.9 mm, *N* = 20). Frontonasal in contact with the rostral and loreals, separating the nasals, latter slightly swollen; nostril—with a large inner flap attached posteriorly—situated in the posterior part of the nasal and in contact with the loreal and first supralabial; frontal separated from the frontonasal by a pair of prefrontals (*N* = 4, topotypes). Scales at anterior edge of ear opening 4-6 on either side of head, projecting outwards as flattened, somewhat spatulate spines, the lowermost one narrow and slender, the one above it distinctly spatulate and the largest; scales generally elongate and somewhat spinose (more so than in *S. swazicus* and *S. warreni*), but short and blunt in TM 55789, 73292 and three juveniles (TM 4275, 26643, 73286). Preoculars 1; supraoculars 4; supraciliaries usually 4 (5 on one side in three specimens); loreals 1; suboculars usually 4–5, occasionally 6 on one side (often different on either side of head), but 3 on right side in NMB R9196); supralabials (anterior to median subocular) usually 4 (3 on one side in three specimens, and 5 on one side in two specimens); infralabials usually 5-6 (5 on left and 7 on right in TM 73281, and 6 left and 7 right in TM 55787); sublabials usually 5 (6 in NMB R9192); occipitals usually 6 (7 in TM 4468 and 4472, 8 in TM 73284; NMB R9192 has a single tiny median granule, TM 73293 has two such granules), outer occipitals shorter than those adjacent to them, and those of the median pair the shortest (*N* = 4); gulars 23-31, but 20 in NMB R9193; dorsal scale rows transversely 28-34, longitudinally 20-24; ventral scale rows transversely 25-28, longitudinally usually 14 (16 in TM 73283); femoral pores per thigh 8-11 (males, *N* = 11), 8-12 (females and juveniles, *N* = 15); differentiated femoral scales (generation glands) in males ≥100 mm SVL: 16-36 per thigh (*N* = 10), in juvenile males <66 mm SVL: 18-22 per thigh (*N* = 2); lamellae under fourth toe 15-19.

*Colour*: (Figs. 2, 12B and 12C) Back dark brown to black with cream to yellow markings (mostly transversely enlarged) forming 4-5 (usually 4, as in holotype TM 4273, Fig. S1) interrupted transverse bands (usually with slightly dark borders in preserved material) that continue onto the tail, together with a band on the nape and a cream spot, blotch or elongate marking (absent in NMB R9193 and TM 73286) immediately behind the median

occipitals. Belly mostly brown (older preserved material; including the holotype, Fig. S2) to black (as in life) with a few cream patches or short 'bands' on either side, which are occasionally joined to the pale bands on the side of the back. The flanks are dark brown to black, usually with narrow, cream to yellowish, vertically elongated bars, occasionally spots (e.g. TM 51066). Top of limbs with numerous irregular cream to yellowish spots and blotches; underparts of limbs mostly cream with irregular brown markings, occasionally bands. Top of head brown or black with scattered irregular cream markings. Throat (including sublabials) mostly black or brown with occasional irregular scattered cream markings, but about half dark and half pale in TM 73283 and 73293, and mostly plain cream in TM 4275 (juvenile).

*Cranial skeleton, postcranial skeleton and dermal osteology* (Fig. 11)
The cranial skeleton, postcranial skeleton and osteoderms are all similar to those described for *S. swazicus* sp. nov. However, in *S. barbertonensis* ventral osteoderms were absent, and quadrates lacked a pronounced ridge and concave region at the lateral edge of the posterior origin of the *adductor musculus mandibulae* (quadrates have a pronounced ridge and concave region in *S. swazicus* sp. nov.).

**Size.** Largest male (TM 73287, Broedershoek) 129.0 + 182.8 = 311.8 mm, but TM 51066 (between Karino and White River) has SVL of 134.0 mm (tail broken/missing). Largest female (TM 4273: holotype, Barberton) 134.0 + 175 = 309 mm, but TM 4468 (Barberton) has SVL of 139.9 mm (tail broken/missing).

**Natural history.** Diurnal and rupicolous, living in deep, horizontal (or gently sloping) crevices in and between large granitic boulders, often in the partial shade of trees (Fig. 12A; see also *Jacobsen, 1989*). For *S.* '*barbertonensis*', *FitzSimons (1943)* noted that the diet is similar to that of *S. warreni* (see below), but includes cetonid beetles and small land snails; usually five young are produced, and based on his examination of a series of females, fertilisation occurs in early spring, with young born at the end of summer. However, one of *FitzSimons' (1943)* localities ('Hluti-Goedgegun', Eswatini) is within the range of *S. swazicus* sp. nov., so it is not possible to know which species his data applies to. Computed Tomography scanning of NMB R9192 revealed four large embryos, and a large beetle in the stomach.

**Distribution.** Restricted to the Barberton, Nelspruit and Khandizwe areas of eastern Mpumalanga Province, South Africa (Fig. 9) at elevations of 724 to 1,008 m a.s.l. An isolated record for this species at Farm: Jessievale, Ermelo district (2630AB, *Bates et al., 2014*) is in fact referable to *Cordylus vittifer* (re-examination by both authors of VM no. 1400 on ReptileMAP).

**Localities.** SOUTH AFRICA: Mpumalanga Province. Barberton (25°47′S, 31°03′E) TM 4273–5, 4468–9, 4471–2; Barberton army base—NMB R9191, 9196 (25°46′26″S, 31°03′20″ E; 861 m a.s.l.), NMB R9192–3 (25°46′27″S, 31°03′21″E; 861 m a.s.l.); Barberton Townlands (25°47′S, 31°03E) TM 73281, 73283–4; Broedershoek 129JU (25°27′S, 31°07′E; 753 m a.s.l.) TM 73286–7; Friedenheim 282JT (25°26′S, 30°59′E; 754 m a.s.l.)

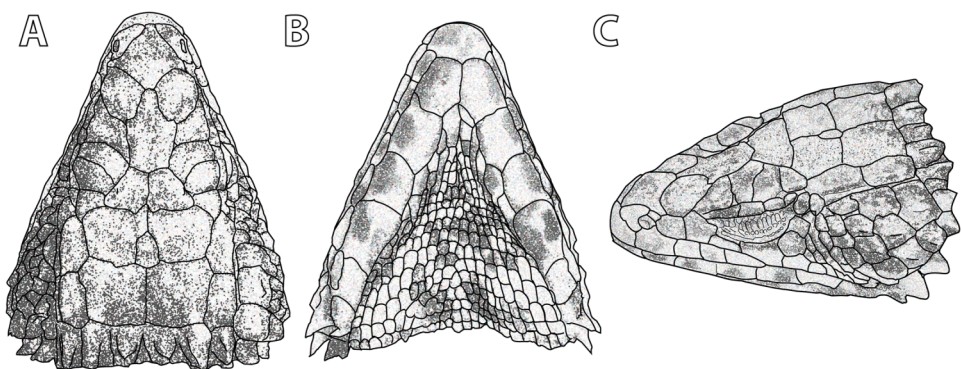

**Figure 13** *Smaug warreni*. **(A) Dorsal, (B) ventral and (C) lateral views of the head of TM 50130 (Lomahasha, Eswatini).** (Drawing credit: E.L. Stanley).

TM 55787; Karino and White River, between (2531AC) TM 51066; Khandizwe (25°28′S, 31°25′E; 724 m a.s.l.) TM 73292–3; Nelspruit, 5 km S of (25°32′S, 30°57′E; 824 m a.s.l.) TM 44873; Nelspruit, 14 km W of, on road to Machadodorp (2530BD) TM 26643; Nelspruit, 82 Ehmke Street, Extension 5 (25°29′37.1″S, 30°59′29.7″E; 793 m; Fig. 12); Nelspruit, Van Riebeeck Park (25°28′S, 30°59′E) TM 55788–9.

**Notes.** *Van Dam's (1921)* type locality of 'Barberton' does not indicate the exact location at which the specimens were collected, so it is considered appropriate to treat both 'Barberton Townlands' (*Jacobsen, 1989*) and 'Barberton army base' as topotypic.

***Smaug warreni*** (*Boulenger, 1908*)
Lebombo Dragon Lizard
Figures 13–15

*Zonurus warreni Boulenger, 1908*: 232 (Ubombo) Syntype: NHM 1946.8.8.1 (see *Reissig, 2014*: 187); *Hewitt, 1909*: 36; *Boulenger, 1910*: 467 & 468; *Power, 1930*: 14 & 17; *FitzSimons, 1930*: 30; *Lawrence, 1937*: 111.
*Cordylus warreni warreni FitzSimons, 1943*: 424 (Ubombo & Ingwavuma); *Loveridge, 1944*: 19 (Ubombo); *Branch, 1988*: 164, *1998*: 195; *Jacobsen, 1989*: 586 (Duikershoek, Halfkroonspruit, Jozini Dam, Mananga, The Hippos); *Adolphs, 1996*: 15; *Bourquin, 2004*: 96 (KwaZulu–Natal); *Adolphs, 2006*: 22.
*Cordylus warreni Alexander & Marais, 2007*: 261 (but photograph on p. 259 is of a *Smaug depressus*, see same photograph in *Bates et al. (2014)*: 212, left photo); *Parera, Ratnayake-Parera & Procheş, 2011*: 14.
*Smaug warreni warreni Stanley et al., 2011*: 64; *Bates et al., 2014*: 210; *Reissig, 2014*: 187.
*Smaug warreni Stanley & Bates, 2014*: 905; *Mouton et al., 2018*: 463.

**Diagnosis.** Distinguished from all other cordylids by its unique combination of dorsal, lateral and ventral colour patterns (see descriptions and figures).

 A medium to large species of *Smaug* distinguishable by the following combination of characters: (1) back usually sandy brown with 5–6 bands (usually interrupted) between

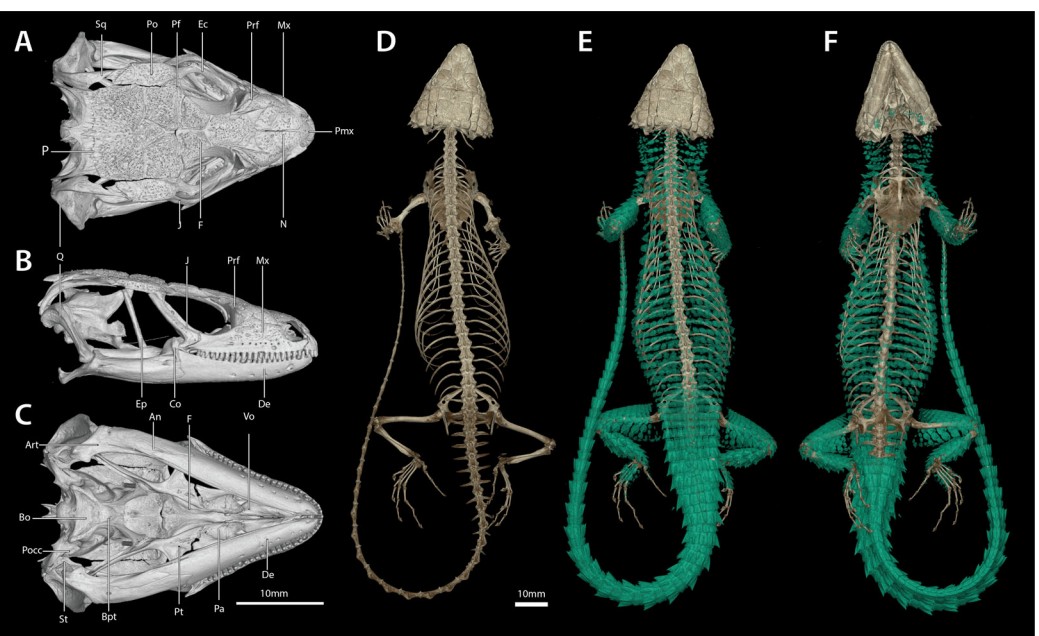

**Figure 14 Cranial (A–C), postcranial (D), and dermal (E, dorsal and F, ventral) osteology of *Smaug warreni* (NMB R9292).** Body ostoderms are highlighted in blue-green. An, angular; Art, articular; Bo, basioccipital; Bpt, basipterygoid; Co, coranoid; De, dentary; Ec, ectopterygoid; Ep, epipterygoid; F, frontal; J, jugal; Mx, maxilla; N, nasal; P, parietal; Pa, palatine; Pf, postfrontal; Pmx, premaxilla; Po, postorbital; Pocc, paraoccipital; Prf, prefrontal; Pt, pterygoid; Q, quadrate; Sq, squamosal; St, supratemporal; Vo, vomer. (Images produced by: E.L. Stanley).

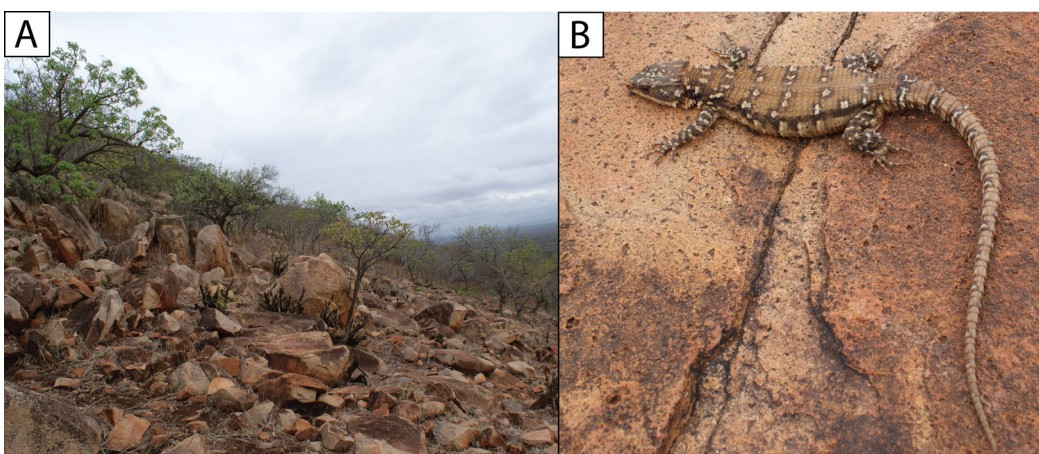

**Figure 15 (A) *Smaug warreni* habitat: north-western slopes of Mount Mananga, northern Lebombo Mountains, Mpumalanga Province, South Africa. (B) Dorsal colouration of *S. warreni*.** (Photo credits: E.L. Stanley).

fore- and hindlimbs, each band consisting of pale, dark-edged markings; (2) pale band on nape behind occipitals; (3) flanks pale with brown markings; (4) belly with brown (often pale) markings on most scales; (5) throat usually mostly pale with scattered small brown spots; (6) usually six enlarged, moderately to non-spinose, occipital scales, middle

pair the smallest, outer occipitals usually equal in length to the adjacent inner ones; an additional small median occipital is often present; (7) dorsolateral and lateral scales weakly spinose; (8) tail moderately spikey; (9) dorsal scale rows transversely 31–41; (10) dorsal scale rows longitudinally 22–28; (11) ventral scale rows transversely 23–27; (12) ventral scale rows longitudinally 14 (rarely 12 or 13); (13) femoral pores per thigh 7–13; subdigital lamellae on 4th toe 15–20.

It differs from other species of *Smaug* as described above in the diagnosis of *S. swazicus* sp. nov. (but maximum SVL in *S. warreni* is 141 mm, and femoral pores in males are 7–13).

Most similar to *S. swazicus* sp. nov. and *S. barbertonensis* but easily distinguished by its colour pattern (see comparisons in diagnosis of *S. swazicus* sp. nov. above); by usually having shorter and blunter scales at the edges of the ear openings compared to *S. barbertonensis*; and quadrates without a pronounced ridge and concave region at the lateral edge of the *adductor musculus mandibulae* posterior origin (with a pronounced ridge and concave region in *S. swazicus* sp. nov.). Also differs as follows: outer occipitals and scales adjacent to them of about equal length (outer occipitals usually shorter than the adjacent inner ones in the other two species); head narrower than *S. barbertonensis* (head width/head length = 73–83% vs. 80–92% in adults); generally higher numbers of transverse dorsal scale rows (32-38 in 92% of specimens) than *S. barbertonensis* (29-32 in 81%); and greater numbers of longitudinal dorsal scale rows (22-28, mean 23.6) than *S. barbertonensis* (20-24, mean 21.7).

**Variation.** *External morphology* (*N* = 39 unless otherwise indicated; Fig. 13): Tail length/SVL 106–143% (SVL: 105.0–128.9 mm, *N* = 6); head width/SVL = 22.2–24.2% in males (SVL: 105.3–127.3 mm, *N* = 9), 19.4–23.3% in females (SVL: 100.6–141.1 mm, *N* = 21); head width/head length = 72.5–83.1% (SVL: 100.6–141.1 mm, *N* = 30); head depth/head length = 35.0–47.2% (SVL: 100.6-141.1 mm, *N* = 31). Frontonasal in contact with the rostral and loreals (often in narrow contact, especially on right side in TM 50130), separating the nasals, latter slightly swollen; nostril—with a large inner flap attached posteriorly—situated in the posterior part of the nasal and in contact with the loreal and 1st supralabial (TM 50130: loreal in very narrow contact with nostril on left side of head, and separated from nostril by upward prolongation of first supralabial on right; posterior part of both supranasals separated by a suture to form a small rectangular scale) (*N* = 16). Frontal usually separated from the frontonasal by a pair of prefrontals, but in broad contact in TM 47449, 50130 (Fig. 13), 50660–1 (*N* = 16), and NMZB-UM 30514 (D.G. Broadley, 2014, personal communication). Scales at anterior edge of ear opening usually 4–5 (3 on right side of head of TM 63567) projecting outwards as flattened, somewhat spatulate, spines, the lowermost one narrow and slender (especially so in TM 47449), the one above it distinctly spatulate and the largest; middle scales usually somewhat blunt and rounded, but generally long and somewhat spinose in TM 47449, 50130, 53869, 70961 (*N* = 16). Preoculars 1; supraoculars 4 (2nd and 3rd on left side largely fused in NMB R10913); supraciliaries usually 4-5 (3 on left side of TM 78967, 4 left and 6 right in TM 13639, 6 on left side of 78969); loreals 1; suboculars 4-5, but 3 on left side

of NMB R10911; supralabials (anterior to median subocular) usually 4 (3 on one side in four specimens, 5 on one side of TM 78967); infralabials usually 6 (5 on both sides of TM 582, 5 on one side of TM 47449, 7 on both sides of TM 15320, 7 on one side in six specimens); sublabials 5; occipitals usually 6, but an additional—often narrow and much elongated—scale medially in 41% of specimens (median scale large in TM 50660, granular in TM 78966); outermost occipitals and those adjacent to them of similar size and length, but scales of the inner (middle) pair shorter and often smaller, except in NMB R10878 in which all occipitals are of similar length, although those of the inner pair are wider ($N = 16$); gulars 23–32 ($N = 37$); dorsal scale rows transversely 31–41, longitudinally 22–28; ventral scale rows transversely 23–27, longitudinally usually 14 (13 in NMB R9199, 12 in TM 2808 and 78969); femoral pores 7–13 (males (smallest is 86.2 mm SVL), $N = 11$), 8–13 (females and juveniles, $N = 27$); differentiated femoral scales (generation glands) in males 13–38 per thigh ($N = 10$); lamellae under fourth toe 15–20.

For the seven specimens examined by D.G. Broadley (2014, personal communication; see below): occipitals 6; dorsal scale rows transversely 34–38, longitudinally 22–24 (26 in NMZB–UM 1542); femoral pores/thigh 9–13.

*Colour*: (Figs. 2 and 15B) Back usually sandy brown with irregular, black-bordered, cream or white blotches (ocelli) forming 5–6 (seven in NMB R10911) slightly to greatly interrupted transverse bands between the legs that continue onto the tail, together with a band immediately behind the occipitals and another on the nape ($N = 16$). This colour pattern is often evident in live (Fig. 15B) and preserved specimens (Fig. 2, specimens preserved for over 30 years). Ocelli may be in close proximity and even set within a continuous black band. Occasional specimens have grey-brown backs with small, widely separated white spots lacking obvious black-borders (e.g. fig. 212 in *Reissig, 2014*). All six NMB specimens from Manyiseni region in KwaZulu–Natal (preserved for over 12 years) have medium brown backs and the pale markings have only moderately distinct borders. Belly white to cream, usually with numerous small, square, rectangular or irregular pale to dark brown markings; occasionally mostly without markings except for the sides (e.g. NMB R10898 and 10912) or with a large dark blotch on each ventral (NMB R9292). The flanks are mostly pale whitish, occasionally with some darker colouring and pale vertical bars. Top of limbs with numerous irregular cream to yellowish spots and blotches; underparts of limbs mostly cream with occasional scattered, irregular brown markings. Top of head tan/khaki brown with dark brown patches and scattered, irregular, cream markings (or small yellow speckles or blotches, observed in photographs of live specimens; also *Ping, 2019*); throat mostly white to cream with varying amounts of darker markings in the form of small spots and blotches (often extensive and bold (e.g. TM 53869), but less so than in *S. swazicus* sp. nov.).

*Cranial skeleton, postcranial skeleton and dermal osteology* (Fig. 14)
The cranial skeleton, postcranial skeleton and osteoderms are all similar to those described for *S. swazicus* sp. nov. However, the quadrates of *S. warreni* lack a pronounced ridge and

concave region at the lateral edge of the posterior origin of the *adductor musculus mandibulae* (quadrates have a pronounced ridge and concave region in *S. swazicus* sp. nov.).

**Size.** Largest male (TM 78963, Mananga Mountain, Mpumalanga, South Africa) 127.3 + 193.5 (on museum tag) = 320.8 mm. Largest female (NMB R9292, Mananga Mountain) 128.9 + 182 = 311 mm, but TM 53869 (Lomahasha, Eswatini) has SVL of 141.1 mm (tail broken/missing).

**Natural history.** Diurnal and rupicolous, occurring in crevices between or under rocks on outcrops along the Lebombo Mountains (Fig. 15A). According to *FitzSimons (1943)*, ants, beetles, fossorial wasps, myriapods, frogs and lizards are eaten. *Loveridge (1944)* noted that for a sample from Ubombo, one specimen had eaten 32 *Eristalis* (drone fly) maggots, another lizard contained millipedes and ants, while a third had consumed a large grasshopper. Females give birth to 4–5 young in late summer (*FitzSimons, 1943*).

**Distribution.** Endemic to the Lebombo Mountains of eastern Eswatini, adjacent western Mozambique and South Africa (north-eastern Mpumalanga and north-eastern KwaZulu–Natal) (Fig. 9) at elevations of 103 to 697 m a.s.l.

**Localities.** (*specimens examined by D.G. Broadley—not included in morphological analysis, but scalation data noted above) MOZAMBIQUE: Estatuene (26°24′18″S, 32°04′42″E) NMZB-UM 30510-3*; Meponduine (25°56′45″S, 31°58′44″E) NMZB 30514*, 30562*; 12 km SE of Komatipoort, close to Komati River (25°28′27.6″S, 32°04′01.0″E; 103 m; photo: L. Verburgt, 2015, personal communication; photo in *Reissig, 2014*: 188, fig. 213). SOUTH AFRICA: KwaZulu-Natal Province. Bhokweni (27°22′S, 32°03′E) TM 78969–72, 78974; Ingwavuma (27°08′S, 32°01′E) TM 15319–20; Manyiseni region, near Mayaluka (26°54′52″S, 32°00′31″E; 642 m) NMB R10878, 10910–3; Manyiseni region, near Mabona (26°52′35″S, 32°04′41″E; 302 m) NMB R10898; Farm: Middlein 84 (27°21′03.0″S, 31°59′10.7″E; 697 m; photographic record: S. Nielsen); Ubombo (27°34′S, 32°05′E) TM 582, 2808, 13639–41; NMZB-UM 1542*. Mpumalanga Province. Farm: Duikerhoek 489JU (25°42′S, 31°57′E) TM 78966; Halfkroon Spruit, Kruger National Park (2531BD) TM 78973, 'J6955' (not examined); Mananga Mountain (25°58′S, 31°52′E) TM 78961–5; Mananga Mountain, 2 km SSW of Nsizwane (25°54′21″S, 31°52′21″E; 347 m) NMB R9197–200, 9292; Farm: The Hippos 192JU (25°28′00″S, 31°57′30″E) TM 78967–8. ESWATINI: Lomahasha (25°59′S, 31°59′E) TM 50130–1, 53869, 63567; Lubombo foothills, 6 km E of Big Bend (26°47′S, 31°59′E) TM 70960–1; Siteki, S of (26°28′S, 31°56′E) TM 47449; Tshaneni (25°59′S, 31°46′E) TM 50660–1. NO DATA: AMNH 173381 (used for CT scanning).

**Notes.** *Boulenger's (1908)* description was based on two male specimens (i.e. syntypes) from Ubombo in the Lebombo Mountains of KwaZulu-Natal. His description includes a plate with a splendid drawing (by A.H. Searle) depicting a specimen with somewhat indistinct, narrow, dark crossbars on the back, each containing scattered pale spots. This illustration, together with *Boulenger's (1908*: 233*)* description: 'Dark brown above, with small yellow black-edged spots forming more or less regular transverse series on the body;

lower parts pale brown', characterise *S. warreni* (but back usually light brown, see Fig. 15B). *Reissig (2014)* noted that the 'type specimen' is NHM 1946.8.8.1.

## A revised diagnostic key to the genus *Smaug*

1a. Occipitals greatly enlarged, the outer ones strongly spinose and about twice as long as those of the median pair; dorsal scales strongly spinose; ventral scales imbricate; lamellae under fourth toe 10–12 .................................................................. *S. giganteus*

1b. Occipitals moderately to weakly enlarged, those of the outer pair somewhat larger or of similar size to the others; dorsal scales not strongly spinose; ventral scales non-imbricate; lamellae under fourth toe 14–20 ..................................................................... 2

2a. Occipitals of the outermost pair largest (and longest), innermost the smallest (and shortest) ................................................................................. 3

2b. Occipitals of the outermost pair not the largest (or longest), innermost of similar size to other occipitals or slightly smaller (and shorter) ...................................... 4

3a. Dorsum mostly plain brown, at most with occasional scattered pale markings; belly cream or brown; throat plain or with small brown spots; ventrals in 10–14 rows longitudinally ..................................................................... *S. breyeri*

3b. Dorsum brown with transversely enlarged cream markings, at least at the sides of the back, but often extensively on the back and tail; belly dark with short pale transverse markings, especially towards the edges; throat pale with dark reticulations; ventrals in 12–16 rows longitudinally ..................................................... *S. vandami*

4a. Back with few or no pale markings; flanks mostly plain, and brightly coloured (red, orange or yellow) in males; first supralabial with distinct upward prolongation; dorsals in 22–30 rows longitudinally ......................................................... 5

4b. Back usually with distinct pale markings (except the 'laevigata' form of *S. depressus*); flanks of males with light and dark markings, and not brightly coloured; first supralabial with moderate or no upward prolongation; dorsals in 13–28 rows longitudinally ...... 6

5a. Loreal large and not elongated, separated from nostril by upward prolongation of first supralabial; preocular usually widely separated from the nasal by the loreal; throat of male uniform dark brown ................................................... *S. mossambicus*

5b. Loreal small and elongated, in contact with nostril; preocular large and usually in contact (or nearly so) with the nasal above the loreal; throat of male yellow or orange, with dark infuscations.................................................................. *S. regius*

6a. Back usually with distinct, small to moderate, scattered white spots or irregular markings, not forming crossbands, or completely plain grey ('laevigata' form); dorsals in 13–21 rows longitudinally ........................................... *S. depressus*

6b. Back with distinct crossbands (usually interrupted) consisting of pale spots, blotches or transversely enlarged bars, often with dark edges; dorsals in 20–28 rows longitudinally .................................................................................7

7a. Outer occipital of similar length to the occipital adjacent to it; small scale often present between median occipitals; back medium to light brown with pale, usually distinctly dark-edged, spots or blotches forming crossbands; belly usually with centre of each scale brown (not mostly black or brown, or with brown crossbars interrupted by a median band); throat with small brown spots...................................... *S. warreni*

7b. Outer occipital usually shorter than the occipital adjacent to it; usually no small scale present between median occipitals; back dark brown to black with pale markings, mostly in the form of narrow, transversely enlarged bars forming crossbands; throat black or with dark reticulations ................................................................... 8

8a. Back with 4–5 pale crossbands between the fore-and hindlimbs, with a pale spot on the nape behind the occipitals; throat mostly black; flanks dark with narrow, pale, vertical bars; belly mostly black, with a few pale markings at the sides; scales at anterior edges of ear openings often elongated and spinose; dorsals in 28–34 (mostly ≤ 32) rows transversely ....................................................... *S. barbertonensis*

8b. Back usually with 5–6 pale crossbands between the fore-and hindlimbs, with a pale band on the nape behind the occipitals; throat pale with dark reticulations; flanks with large cream spots and blotches; belly with brown crossbars interrupted by a dark median band; scales at anterior edges of ear openings usually short and blunt; dorsals in 31–41 (mostly ≥ 32) rows transversely ..................................... *S. swazicus* sp. nov.

## DISCUSSION

Examination of voucher specimens (NMB) used for the molecular analysis of *Stanley & Bates (2014)*, as well as most other available museum material of the three lineages, indicated that the 'Eswatini' lineage—including populations in a small area on the northern Eswatini-Mpumalanga border, and northern KwaZulu–Natal Province in South Africa—was readily distinguishable from *S. barbertonensis* sensu stricto (and *S. warreni*) by its unique dorsal, lateral and ventral colour patterns. *FitzSimons (1943)* had in fact noted differences in colour pattern between specimens of *S. barbertonensis* from the type locality of Barberton and specimens from 'Hluti–Goedgegun' in Eswatini (now referred to the new species), but this had been regarded as merely representing regional variation.

Multivariate analyses of scale counts and body dimensions indicates that the 'Eswatini' lineage and *S. warreni* are most similar. In particular, *S. barbertonensis* differs from the other two lineages by its generally lower numbers of transverse rows of dorsal scales, more spinose scales at the anterior edges of the ear openings, and a relatively wider head. Also, the outer and adjacent inner occipital scales in *S. warreni* are of similar length, distinguishing it from the other two species which have the outer occipital usually slightly shorter that the adjacent inner occipital. High resolution Computed Tomography reveals differences in cranial osteology between specimens from all three lineages, with the 'Eswatini' lineage being remarkable in having a pronounced ridge and concave region at the lateral edge of the posterior origin of the *adductor musculus mandibulae*.

The 'Eswatini' lineage is described here as *Smaug swazicus* sp. nov., on the basis of genetic distinctiveness and diagnostic scalation and colour patterns. It appears to have a fairly widespread distribution in Eswatini west of the Lebombo Mountains, and a somewhat peripheral distribution in South Africa near Eswatini's borders with Mpumalanga and KwaZulu–Natal provinces. We estimate that about 90% of this species' range is in Eswatini and suggest that it be considered near-endemic to that country. Recognition of the new species means that *S. barbertonensis* sensu stricto is a South African endemic restricted to an altitudinal band of about 300 m in the Barberton–Nelspruit–Khandizwe area of eastern Mpumalanga Province, while *S. warreni* is endemic to the narrow Lebombo Mountain range of South Africa, Eswatini and Mozambique.

The phylogenetic analysis of *Stanley & Bates (2014)* did not include samples from the southern part of the range (especially KwaZulu–Natal) of *S. swazicus* sp. nov., but northern (including one locality in Mpumalanga) and southern material is morphologically indistinguishable, and so we provisionally treat all these populations as *S. swazicus* sp. nov.

The geographical break between *S. barbertonensis* and *S. swazicus* sp. nov. appears to correspond to the location of the ancient Makhonjwa Mountain range, which lies directly south of Barberton. This range, also referred to as the Barberton Greenstone Belt, is made up of some of the world's oldest exposed rocks (3.6 billion years old) which contain fossilised evidence of the earliest life on Earth (*De Wit, 2010*). The time-calibrated phylogenetic analysis of the *S. warreni* species complex by *Stanley & Bates (2014)* indicates that the *S. warreni*-*S. swazicus* sp. nov. lineage diverged from *S. barbertonensis* during the late Miocene, around 7.5 million years ago. This is somewhat earlier than the most recent and extreme period of uplift of the eastern escarpment (*Partridge & Maud, 1987*), suggesting that populations on either side of the Makhonjwa Mountains were isolated before that time. The population east of the Makhonjwa Mountains split around 6.2 million years ago (*Stanley & Bates, 2014*), after which time *S. warreni* became closely associated with the narrow Lebombo Mountain range.

The distribution/endemicity pattern seen in the *S. warreni* species complex is approximated by the southern-most taxa of rupicolous flat lizards in the *Platysaurus intermedius* Smith, 1844 species group (see *Scott, Keogh & Whiting, 2004*; *Bates et al., 2014*): the range of *P. i. wilhelmi* *Hewitt, 1909* approximates that of *S. barbertonensis* (although the former also occurs further north), that of *P. i. natalensis* FitzSimons is similar to *S. swazicus* sp. nov., and *P. lebomboensis* Jacobsen is, like, *S. warreni*, restricted to the Lebombo Mountain range (Fig. 8). A taxonomically comprehensive phylogenetic analysis of *Platysaurus* that will clarify relationships and make further geographic comparisons possible is in preparation (S. Keogh, 2019, personal communication). In the *Afroedura multiporus* (Hewitt) group of rupicolous flat geckos (see *Jacobsen et al., 2014*; *Bates et al., 2014*), *A. haackei* (Onderstall) has a similar distribution to *S. barbertonensis*, and appears to be separated from *A. major* (Onderstall) in north-western Eswatini by the Makhonjwa range, as are *S. barbertonensis* and *S. swazicus* sp. nov. (Fig. 9). In the thread

snake genus *Leptotyphlops* Fitzinger, *L. telloi* Broadley & Watson is also endemic to the Lebombos (*Bates et al., 2014*).

## Conservation implications

Due to their obligate rupicolous ecology, members of the *Smaug warreni* species complex are not subject to the same levels of habitat destruction as their terrestrial congener, *S. giganteus.* Jacobsen (1989) listed all species of *Smaug* (except the Vulnerable *S. giganteus*) as protected schedule 2 (Transvaal Nature Conservation Ordinance 12 of 1983), while *Bates et al. (2014)* and *Bates & Mouton (2018a, 2018b)* reported the global conservation status of these species as 'Least Concern', while recommending that further research is needed to assess the impact of tree removal from the habitat of *S. barbertonensis* (i.e. *S. barbertonensis* and *S. swazicus* sp. nov.) as crevices in the partial shade of trees are often selected for shelter (*Jacobsen, 1989*). In this regard, R.C. Boycott (2019, in litt.) noted that when he visited the locality 'between Hluti and Goedgegun' in Eswatini (as reported by *FitzSimons (1943)*) a few years ago, *S. swazicus* sp. nov. was not present, possibly because all large trees along the rocky hillsides had disappeared, such that dappled shade was no longer available. The species appeared to have been replaced by skinks (*Trachylepis varia* (Peters) and *T. margaritifer* (Peters)). Part of the natural range of *S. swazicus* sp. nov. was inundated and thus lost to the species when the Maguga Dam in Eswatini was filled in 2002/3, although about 20 specimens were collected by Boycott and relocated downstream from the Dam as part of the Maguga Dam Comprehensive Mitigation Plan (R.C. Boycott, 2019, in litt.). The recognition of *S. swazicus* sp. nov. means that the range of *S. barbertonensis* sensu stricto now covers only about 180 km$^2$. Also, this species has been recorded within a narrow altitudinal band of only 300 m. Its conservation status should therefore be monitored. Despite being endemic to the narrow Lebombo mountain range, *S. warreni* apparently does not face any significant threats, and it is therefore also considered 'Least Concern' (*Bates et al., 2014*; *Bates & Mouton, 2018b*). It appears to occur throughout the Lebombo range, from low to high altitudes (103 to 697 m a.s.l.). Using *IUCN (2012, 2017)* criteria, we suggest that all three species be regarded as Least Concern at this time.

## CONCLUSIONS

Following the finding by *Stanley & Bates (2014)* that the south-eastern assemblage of populations referable to the *S. warreni* species complex comprised three distinct genetic lineages, we hypothesised that morphological differences should also exist between specimens referable to these lineages. Distinct differences were indeed identified between populations with regard to colour pattern, scalation and cranial osteology, necessitating the description of a new species, *S. swazicus* sp. nov., which appears to be near-endemic to Eswatini. This finding means that *S. barbertonensis* sensu stricto is endemic to South Africa, with a restricted range that may require monitoring in future to ensure that the species does not become threatened with extinction. Also, sampling of populations referable to *S. swazicus* sp. nov. in South Africa's KwaZulu–Natal Province is needed to investigate whether additional cryptic diversity exists in this species complex. *Smaug warreni*

is endemic to the Lebombo range in South Africa, Eswatini and Mozambique. There are now nine known species of dragon lizards (*Smaug*).

# ACKNOWLEDGEMENTS

We thank Lauretta Mahlangu (Ditsong Natural History Museum, Pretoria) for access to, and for loans of, *Smaug* material in her care; the late Donald Broadley for data on specimens in the collection of the Natural History Museum of Zimbabwe (Bulawayo); Richard Boycott for information about the distribution of this genus in Eswatini; and Theo Busschau for the use of his photographs of *S. barbertonensis* and its habitat.

### Funding

The authors received no funding for this work.

### Competing Interests

The authors declare that they have no competing interests.

### Author Contributions

- Michael F. Bates conceived and designed the experiments, examined the specimens, analysed the data, prepared figures and/or tables, authored or reviewed drafts of the paper, and approved the final draft.
- Edward L. Stanley conceived and designed the experiments, conducted the CT scanning, analysed the data, prepared figures and/or tables, authored or reviewed drafts of the paper, and approved the final draft.

### Animal Ethics

The following information was supplied relating to ethical approvals (i.e. approving body and any reference numbers):

The National Museum Bloemfontein Ethics Clearance Committee provided full approval for this research (#NMB ECC 2019/13).

### Data Availability

Morphological data for specimens of *Smaug warreni*, *S. barbertonensis* and *S. swazicus* sp. nov. examined for this study are available in the Supplemental Files.

All specimens examined are listed (with museum numbers) in Table S2 and are housed at the National Museum (Bloemfontein) (NMB) or Ditsong National Museum of Natural History (Pretoria) (TM). Specimens AMNH 173382 (*Smaug swazicus* sp. nov.) and AMNH 173381 (*Smaug warreni*) were used for skeletal analysis only and are housed at the American Museum of Natural History, New York (AMNH).

CT-scanned specimens are available at MorphoSource: https://www.morphosource.org/Detail/ProjectDetail/Show/project_id/828.

## New Species Registration

The following information was supplied regarding the registration of a newly described species:

Publication LSID: urn:lsid:zoobank.org:pub:490BDD66-155F-423F-A4E9-DEAEEB024CC5

Squamata Cordylidae. Bates and Stanley LSID: urn:1sid:zoobank.org:act:A942675E-5E76-4FC9-AA8F-BFA7A4C131C7.

## Supplemental Information

Supplemental information for this article can be found online at http://dx.doi.org/10.7717/peerj.8526#supplemental-information.

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
