# Peer review of "A taxonomic revision of the south-eastern dragon lizards of the Smaug warreni (Boulenger) species complex in southern Africa, with the description of a new species (Squamata: Cordylidae)"

_PeerJ, doi:10.7717/peerj.8526_

## Round 0.1 · original submission · Minor Revisions

Dear Dr. Bates,

All reviewers find the paper well written and worth of publication. They suggest some minor change to further improve the manuscript. So, I ask you to improve the manuscript according to the few tips of reviewers.
Once again, thank you for submitting your manuscript to PeerJ and we look forward to receiving your revision. Please, respond point-to-point to the comments of reviewers to speed up the process of revision

Best regards
Gabriele Casazza

·

Basic reporting

Excellent with very few mistakes.
References complete.
Well illustrated with clear maps and drawings.

Experimental design

The authors use several museum "acronyms" but do not indicate a bibliographic reference to explain which acronym list they follow - this is important since many museums have several distinct acronyms.
Authors have to precise if colour is a character that can be used on preserved collection specimens and if that character is still obvious in older preserved specimens - this is important since many of their identification characters are based on colour and colouration pattern. Authors have to distinguish between colour pattern observed on live and preserevd specimens or to explain that the colour pattern of live specimens is preserved in collection.
Experimental design of that paper is correct.

Validity of the findings

Findings are robust.
Conclusions are correct.

Additional comments

Muscles have to be written in italic.
You indicate ratio of all characters in percentage but not for TL/SVL.
Check since authors cited as reference in the paper are not all in italic.
Can you not provide some illustrations for two or three morphological characters used in the identification key?
The first part of the discussion is just a repetition of what was previously said and looks more like a summary than a discussion. Discussion could include some references about other animal or vegetal groups that follow the same distributional and endemicity pattern than the S. warreni group.
Bibliographic references are not homogeneous e.g. the last author of a paper (if several) is not indicated in the same way throuhout the references.
It should be indicated that colouration of the holotype is taken on a more than 10 years preserved specimen not on a live specimen.

Reviewer 2 ·

Basic reporting

This paper looks at a lineage of the Smaug warren complex that was found to be distinct in a previous molecular phylogenetic study. Using external morphology and ct scans to describe osteology, the authors provide a nice taxonomic work describing a new species for eswatini. The paper is well written, but I have some concerns that should be addresses before this paper is published. First, more details are needed on the molecular phylogenetic evidence that is used to support the hypothesis presented here, so the reader does not have to go to the original paper. Hopefully several independent nuclear markers are used in a multi-species coalescent context, to deal with potential incomplete lineage sorting. Also, the morphological data could be analyses in a more rigorous way, using proper multivariate statistical analysis (e.g. linear models) and possibly species delimitation analyses like iBPP. Below I outline some more minor issues.


1. I find odd putting taxonomic author names but not year of description following species, like in S. warreni and S. barbetonensis in abstract.

2. The type of molecular data used in the referenced phylogeny in the abstract should be mentioned. Was it multi-locus, just mitochondrial, phylogenomic, etc…

3. I think a general broader introduction to girdled lizards is warranted to put the reader in context, rather to jumping straight to the Smaug warren complex. Maybe start with cordylids briefly, then introduce Smaug and then this species complex.

4. Line 55: I am not a native English speaker, but have never heard the term “rupicolous”, and if it means the same as saxicolous, I suggest using that term instead since I think it is more familiar to a broader audience.

5. I think the convoluted taxonomic history of the complex can be well summarised in a table, as a supplement to the text.

6. Line 76: If want to add an alternative spelling to Eswatini, do it upon the first mention of the country

7. Lines 81-92: Definitely more details are needed for this molecular study, in particular what kind of data were used, since it seems to be central to the conclusions of this paper. I would be careful if this is based just on mitochondrial loci, since it has been increasingly shown how susceptible these data is to introgression and incomplete lineage sorting.

8. Line 118: how is TM an abbreviation of a museum with no T word in it?

9. Line 123: missing “the” before “American”

10. Line 137: I think Ethics approval can just go in acknowledgements.

11. Indicate if all measurements were performed by the same person or not.

12. Line 188: “Statistical and multivariate analyses”. Multivariate analyses are also statistical, and first line of the paragraph says “univariate”. Maybe just call it statistical analyses.

13. Line 189: It is weird to say what program you used before describing the analyses.

14. Line 190: first time I read about “mensural characters” (again I admit, I am not a native English speaker, but I do read a lot of taxonomy). Maybe use continuous character.
15. Line 191: state based on what are specimens over 70 mm considered adults. Its just an arbitrary call?

16. The section on statistical analyses needs more detail on what actual analyses where performed. PCA is just an ordination/visualization tool. I think proper multivariate statistics (e.g. MANOVA) are needed to assess differences in morphology between species.

17. It is better for readership is sections of the results match sections of the methods.

18. Line 269-274: I think finding something “slightly more spinose” sounds extremely ambiguous. Can you measure this?

19. PCA shows extensive overlap between the new species and S. warren. These results do not convince me the species have different morphology. I am not saying this means they are not different species, but this indicates stronger support from molecular data is needed.

20. I think a more detailed molecular assessment is needed, the phylogenetic pattern found might be a product of past mitochondrial introgression.

21. Lines 915-917 more detailed analyses would be good. I also sugest trying proper species delimitation methods, like BPP and iBPP (which can handle morphology as well as molecular data).

Experimental design

no comment

Validity of the findings

no comment

·

Basic reporting

This is a well written manuscript describing a new Smaug from southern Africa. The authors do a good job of providing sufficient taxonomic history information and comparing morphologically similar species using both traditional techniques and high resolution CTscans. However, I have a few comments and suggestions that pottential will benefit the paper.

Material and Methods section:
When referring to the coordinates system the authors should provide the formart that they are using through the text as well as the datum.

Systematics section:
I recommend that the figures that are cited right before the chersonymy should be cited in the subsequent text, where appropriate.

In the diagnosis section the authors simply provide a comparison of the new taxon with other species of the genus. While this is needed and important its not sufficient as a diagnosis. This should start with a combination of unique characters that generally describe the species, only after that the comparison with other species should appear. For the case of Smaug warreni the diagnosis 1) does not need the comparative part because the comparison between it and the new species was already been done in the new species diagnosis, 2) and the species is already been described.

Format in the text:
The structure of the article is conform to an accepteable format of "standard sections" with exception of:

line 94 : delete page number from FitzSimons 1943 is irrelevant

line 128: substitute co-ordinates by coordinates

line 146: FitzSimons 1943 should be in italics

line 681: Jacobsen 1989 should be in italics

line 692: Bates et al. 2014 should be in italics

line 971: Boycott, 2019 should be in italics


Please check the Tables and Figures titles they must be in bold.

Missing title on figures 10 & 13, also they must be identified with A and B, and not with left and right term (see PeerJ Standards).

References:
Please check the literature section, some references are not in full agreement with the PeerJ Standards.

e.g.

line 1007
ADOLPHS, K. 1996. Bibliographie der Gürtekechsen und Schildechsen (Reptilia: Sauria: Cordylidae & Gerrhosauridae). Squamata Verlag, Sankt Augustin, 255 pp.

PeerJ
ADOLPHS, K. 1996. Bibliographie der Gürtekechsen und Schildechsen (Reptilia: Sauria: Cordylidae & Gerrhosauridae). Squamata Verlag, Sankt Augustin.

line 1022
BOULENGER, G.A. 1908. On a collection of fresh-water fishes, batrachians, and reptiles from Natal and Zululand, with description of new species. Annals of the Natal Government Museum (Pietermaritzburg) 1:219˗235.

PeerJ
BOULENGER, GA. 1908. On a collection of fresh-water fishes, batrachians, and reptiles from Natal and Zululand, with description of new species. Annals of the Natal Government Museum (Pietermaritzburg) 1:219˗235.

Experimental design

The experimental design is standard and adequate to such type of study.

Validity of the findings

All the data supports the recognition of the new species.

Additional comments

This is a well written manuscript describing a new Smaug from southern Africa. The authors do a good job of providing sufficient taxonomic history information and comparing morphologically similar species using both traditional techniques and high resolution CTscans. However, I have a few comments and suggestions that pottential will benefit the paper.

Material and Methods section:
When referring to the coordinates system the authors should provide the formart that they are using through the text as well as the datum.

Systematics section:
I recommend that the figures that are cited right before the chersonymy should be cited in the subsequent text, where appropriate.

In the diagnosis section the authors simply provide a comparison of the new taxon with other species of the genus. While this is needed and important its not sufficient as a diagnosis. This should start with a combination of unique characters that generally describe the species, only after that the comparison with other species should appear. For the case of Smaug warreni the diagnosis 1) does not need the comparative part because the comparison between it and the new species was already been done in the new species diagnosis, 2) and the species is already been described.

Format in the text:
The structure of the article is conform to an accepteable format of "standard sections" with exception of:

line 94 : delete page number from FitzSimons 1943 is irrelevant

line 128: substitute co-ordinates by coordinates

line 146: FitzSimons 1943 should be in italics

line 681: Jacobsen 1989 should be in italics

line 692: Bates et al. 2014 should be in italics

line 971: Boycott, 2019 should be in italics


Please check the Tables and Figures titles they must be in bold.

Missing title on figures 10 & 13, also they must be identified with A and B, and not with left and right term (see PeerJ Standards).

References:
Please check the literature section, some references are not in full agreement with the PeerJ Standards.

e.g.

line 1007
ADOLPHS, K. 1996. Bibliographie der Gürtekechsen und Schildechsen (Reptilia: Sauria: Cordylidae & Gerrhosauridae). Squamata Verlag, Sankt Augustin, 255 pp.

PeerJ
ADOLPHS, K. 1996. Bibliographie der Gürtekechsen und Schildechsen (Reptilia: Sauria: Cordylidae & Gerrhosauridae). Squamata Verlag, Sankt Augustin.

line 1022
BOULENGER, G.A. 1908. On a collection of fresh-water fishes, batrachians, and reptiles from Natal and Zululand, with description of new species. Annals of the Natal Government Museum (Pietermaritzburg) 1:219˗235.

PeerJ
BOULENGER, GA. 1908. On a collection of fresh-water fishes, batrachians, and reptiles from Natal and Zululand, with description of new species. Annals of the Natal Government Museum (Pietermaritzburg) 1:219˗235.

---

## Round 0.2 · Minor Revisions

Dear Dr. Bates,

The reviewers find the new version of the manuscript much improved. In particular though, they find some possible mistakes in the use of italics and suggest to check how genetically isolated the lineages/species are. If you think this is not useful, you may put in the manuscript a short explanation on the reason why it isn't, please. So, I ask you to perform these few changes before the manuscript acceptance for publication.
Once again, thank you for submitting your manuscript to PeerJ and we look forward to receiving your revision. Please, respond point-to-point to the comments of reviewers to speed up the process of revision

Best regards
Gabriele Casazza

Reviewer 2 ·

Basic reporting

See below

Experimental design

See below

Validity of the findings

See below

Additional comments

I am pleased to see a much-improved version of the manuscript, where most of the reviewers comments were properly addressed. However, I still believe, given that the authors have plenty of molecular and morphological data, that a species delimitation method should be implemented. Also, a genetic Structure analysis, to see how much admixture there is between these potential taxa, will reveal how genetically isolated these lineages are. If they strongly believe this is not suitable, then they should argue it in the manuscript. We are now moving into an era of systematics where lots of species we have considered real based on phylogenetic trees and morphological differences are actually not evolutionarily independent entities, therefore I think it is better to include a more robust suite of analyses to support the taxonomy.

---

## Round 0.3 · accepted · Accept

Dear Dr. Bates,

I am very pleased to say that your paper "A taxonomic revision of the south-eastern dragon lizards of the Smaug warreni (Boulenger) species complex in southern Africa, with the description of a new species (Squamata: Cordylidae) " is accepted for publication in the PeerJ. Congratulations!

Thank you for submitting your work to PeerJ.

Yours sincerely,
Gabriele Casazza